



# Enhanced soot particle ice nucleation ability induced by aggregate compaction and densification

Kunfeng Gao[1,2,3], Franz Friebel[3],  Chong-Wen Zhou[1], and Zamin A. Kanji[3]

[1]School of Energy and Power Engineering, Beihang University, Beijing, China
[2]Shenyuan Honours College of Beihang University, Beihang University, Beijing, China
[3]Department of Environmental Systems Science, Institute for Atmospheric and Climate Science, ETH Zurich, 8092 Zurich, Switzerland

*Correspondence to*: Zamin A. Kanji (zamin.kanji@env.ethz.ch) and Chong-Wen Zhou (cwzhou@buaa.edu.cn)

**Abstract.** Soot particles, acting as ice nucleating particles (INPs), can contribute to cirrus cloud formation which has an important influence on climate. Aviation activities emitting soot particles in the upper troposphere can potentially impact ice nucleation (IN) in cirrus clouds. Pore condensation and freezing (PCF) is an important ice formation pathway for soot particles in the cirrus regime, which requires the soot INP to have specific morphological properties, i.e. mesopore structures. In this study, the morphology and pore size distribution of two kinds of soot samples were modified by a physical agitation method without any chemical modification, by which more compacted soot sample aggregates could be produced compared to the unmodified sample. The IN activities of both fresh and compacted soot particles with different sizes, 60, 100, 200 and 400 nm, were systematically tested by the Horizontal Ice Nucleation Chamber (HINC) under mixed-phase and cirrus clouds relevant temperatures ($T$). Our results show that soot particles are unable to form ice crystals at $T > 235$ K (homogeneous nucleation temperature, HNT) but IN was observed for compacted and larger size soot aggregates (> 200 nm) well below homogeneous freezing relative humidity (RH_hom) at $T <$ HNT, demonstrating PCF as the dominating mechanism for soot IN. We also observed that mechanically compacted soot particles can reach a higher particle activation fraction (AF) value for the same $T$ and RH condition, compared to the same aggregate size fresh soot particles. The results also reveal a clear size dependence for the IN activity of soot particles with the same agitation degree, showing that compacted soot particles with large sizes (200 and 400 nm) are more active INPs and can convey the single importance of soot aggregate morphology for the IN ability. In order to understand the role of soot aggregate morphology for its IN activity, both fresh and compacted soot samples were characterized systematically using particle mass and size measurements, comparisons from TEM (transmission electron microscopy) images, soot porosity characteristics from argon (Ar) and nitrogen ($N_2$) physisorption measurements, as well as soot-water interaction results from DVS (dynamic vapor sorption) measurements. Considering the soot particle physical properties along with its IN activities, the enhanced IN abilities of compacted soot particles are attributed to decreasing mesopore width and increasing mesopore occurrence probability due to the compaction process.



## 1 Introduction

Black carbon (BC) particles are estimated as the second-most important forcing for climate warming only after $CO_2$ (Ramanathan and Carmichael, 2008; Bond et al., 2013). BC particles can influence the radiation balance in the atmosphere directly by scattering or absorbing shortwave radiation and indirectly by acting as cloud condensation nuclei (CCN) or ice nucleating particles (INPs) in the atmosphere to form water droplets or ice crystals (Bond et al., 2013; Jacobson, 2004), thereby changing cloud properties. For example, McGraw et al. (2020) suggested that BC particles influence cirrus cloud formation by acting as INPs and competing with the homogeneous freezing of aerosol solution droplets, and exhibit a large uncertainty in their global net radiative forcing on climate. Recently, Schneider et al. reported even higher $RH_i$ (relative humidity with respect to ice) for homogeneous freezing of sulphate droplets compared to Koop et al. (2000), which implies that soot particles are even more likely to compete with homogeneous freezing based on the new freezing parameterization suggested in Schneider et al. (2021). As aviation emissions emit BC with a great amount of water vapor in contrail plumes at high altitude and cold temperature ($T$) conditions, aviation soot particles can be activated as ice crystals, which potentially regulates cirrus cloud coverage in aviation corridors.

In the cirrus regime, ice crystals can be formed via homogeneous freezing of solution droplets or heterogeneous freezing on the surface of an INP (Cziczo et al., 2013; Lohmann et al., 2020). Homogeneous freezing, requiring low $T$ and high RH, i.e. $RH_{hom}$ (homogeneous freezing relative humidity) conditions, can face competition from heterogeneous freezing at RH < $RH_{hom}$. Because INPs lower the energy barrier for the ice embryo formation and facilitate ice crystal activation (Vali et al., 2015), which can deplete water vapor that would otherwise be needed to reach $RH_{hom}$ levels. Due to a hydrophobic surface and associated low water interaction ability, soot particles have been assumed to be poor INPs as requiring low $T$ and high RH for ice nucleation. DeMott (1990) investigated the ice nucleation (IN) activity of acetylene ($C_2H_2$) soot in an expansion cloud chamber at $T$ < 253 K and suggested that soot particles are able to form ice crystals via immersion mode freezing when $T$ is lower than HNT (homogeneous nucleation temperature) of supercooled liquid water droplets. Möhler et al. (2005) reported that propane ($C_3H_8$) flame soot particles with low organic carbon (OC) content form ice crystals via deposition nucleation at $T$ < HNT. Kanji and Abbatt (2006) studied the ice nucleation activity of $n$-hexane soot with a cold stage facility under RH conditions below water saturation at cirrus cloud $T$ and observed ice crystals only at the instrument threshold  suggesting that $n$-hexane soot can be activated via deposition nucleation but is a poor INP. Mahrt et al. (2018) studied the IN activities of six kinds of soot particle in a continuous flow diffusion chamber under mixed phase and cirrus clouds conditions. The authors found that porous soot particles are able to form ice crystals below $RH_{hom}$ conditions which the authors attributed to pore condensation and freezing (PCF) (Marcolli, 2014) instead of deposition nucleation. On the contrary, nonporous and hydrophobic soot particles only freeze homogeneously at $T$ < HNT (Möhler et al., 2005; Koehler et al., 2009; Mahrt et al., 2018). PCF was also reported by Nichman et al. (2019) and Zhang et al. (2020) in their studies of soot IN activity at $T$ < HNT. Recently, Kanji et al. (2020) and Falk et al. (2021)demonstrated that soot particles do not form ice at $T$ > HNT. This finding



supports the relevance of soot ice crystal activation via PCF mechanism at cirrus conditions and rules out the role of immersion mode freezing.

65

According to Marcolli (2014) and Marcolli et al. (2021), the PCF process occurs following three steps. Firstly, supercooled water condenses into mesopores (2-50 nm) due to the inverse Kelvin effect below water saturation conditions. Next, the supercooled pore water freezes homogeneously at $T <$ HNT or freezes heterogeneously if active sites are available within the pores. Finally, the pore ice grows out of the pore and forms a macroscopic ice crystal. The PCF mechanism emphasizes the role of pore size distribution (PSD) and water-soot surface interaction ability for soot particle ice nucleation (Marcolli, 2014; David et al., 2020; Marcolli et al., 2021). Given the large heterogeneity of soot properties, including chemical characteristics, like chemical composition, surface polarity and water-soot contact angle, as well as physical morphologies, like aggregate mobility size, particle fractal dimension ($D_f$), soot porosity and PSD, the dominating parameter of PCF for soot particles is still to be revealed. Previous studies suggested that the water interaction history of soot particles, e.g. water droplet or ice crystal formation processes, can lead to compacted soot aggregates with enhanced IN abilities (Colbeck et al., 1990; Ma et al., 2013; China et al., 2015; Bhandari et al., 2019; Mahrt et al., 2020b), suggesting soot aggregate size and the porosity characteristics are crucial for its IN activity.

The IN ability of an INP is known to be dependent on its size (Pruppacher and Klett, 1997; Archuleta et al., 2005; Connolly et al., 2009; Mahrt et al., 2018; Nichman et al., 2019) since active sites on INPs promoting IN scale with the particle surface area. In the case of soot particle and PCF, pore structures generated among primary particle networks in the aggregate are important for soot IN and should also scale with the soot aggregate size. Zhang et al. (2020) suggested that there is a PCF size threshold (~ 200 nm) for soot particles addressed in their study at 227 K but depends on the soot type. However, aviation emissions tend to be comprised of soot particles with a number size distribution mode around 100 nm (Bond et al., 2013). For example, the APEX (Aircraft Particle Emissions eXperiment) campaign shows that the majority of soot particles emitted by an aero-engine are ranging from ~ 3 nm to 100 nm and with a geometric number mean diameter (GMD) of ~ 10-35 nm (Wey et al., 2007; Kinsey et al., 2010). Nonetheless, larger size soot particles from contrail ice crystal residuals were detected in field studies. For instance, aircraft contrail released soot particles with a typical size larger than 400 nm were detected in field sampling studies (Twohy and Gandrud, 1998; Petzold et al., 1998). In order to make the IN experimental results both comparable to the literature and relevant to the real atmosphere, soot particles with a mobility size down to 60 nm and up to 400 nm were investigated in this study.

There is some evidence in the literature showing that soot particle IN ability is controlled by its morphology. Nichman et al. (2019) reported that soot samples with lower branching are more IN active than fractal soot particles. Mahrt et al. also (2020b) reported significant enhancement in IN ability of $C_3H_8$ flame soot due to a compacted morphology induced by cloud processing. In a separate study, Mahrt et al. (2020a) observed significantly enhanced soot ice nucleation after aging soot



particles in water and diluted sulfuric acid ($H_2SO_4$) solution, suggesting that the particle structure collapse caused by droplet evaporation contributes to this ice activation promotion. The above studies either used different soot samples to represent different morphologies or potentially introduced chemical modifications upon changing the particle morphology, thus a change

in chemical composition cannot be excluded and the exclusive role of morphology in soot ice nucleation cannot be differentiated. In this study, soot particle morphology was changed only by mechanical stirring preserving the chemical composition to avoid confounding effects of chemical changes.

For the purposes of this study, a unique soot sample preparation method was used to generate fresh and mechanically

compacted soot particles in the size range spanning from 60 to 400 nm. The particle IN activity was measured in a continuous flow diffusion chamber both at mixed phase and cirrus cloud conditions. Systematic measurements to characterize soot particle morphological properties, inclusion of soot aggregate mass and size, microscopic images for single soot aggregate morphology, as well as soot sample PSD, were performed to interpret the corresponding IN experimental results, in addition to soot water interaction abilities analysed from soot water vapor isotherms.

## 2 Experimental methods

The schematic of the experimental setup is shown in Fig. 1. Firstly, fresh soot samples were directly used or prepared as compacted powders depending on the time of physical agitation. Subsequently, soot powders were either aerosolized by a dry dispersion setup (Fig. 2) to generate aerosol samples or used for bulk sample offline characterization measurements. The soot sample PSD was measured by both argon (Ar) and nitrogen ($N_2$) sorption techniques and the soot-water interaction ability was

measured by dynamic vapor sorption (DVS) measurements. Aerosolized soot particles were size selected by a differential mobility analyser (DMA, classifier 3080, with a 3081 column and a polonium radiation source, TSI Inc.) to generate monodisperse aerosol sample flow, with a mobility size of 60, 100, 200 or 400 nm. The monodisperse aerosol sample flow was diluted by a factor of ~ 6 and split for downstream IN experiments and soot particle online characterization measurements, as depicted in Fig. 1. Soot particle IN abilities were measured by the Horizontal Ice Nucleation Chamber (HINC) (Lacher et

al., 2017; Mahrt et al., 2018) based on the continuous flow diffusion chamber technique developed by Kanji and Abbatt (2009). In parallel, the mass and size distribution of soot particles were measured by a centrifugal particle mass analyser (CPMA, Cambustion Ltd., Cambridge, UK) and a scanning mobility particle sizer (SMPS, Classifier 3082, Column 3081, TSI Inc.) system. Additionally, soot aggregates for transmission electron microscopy (TEM) image analysis were collected by the Zurich Electron Microscope Impactor (ZEMI) (Aerni et al., 2018; Mahrt et al., 2020b). The excess aerosol flow was pumped to the

exhaust.





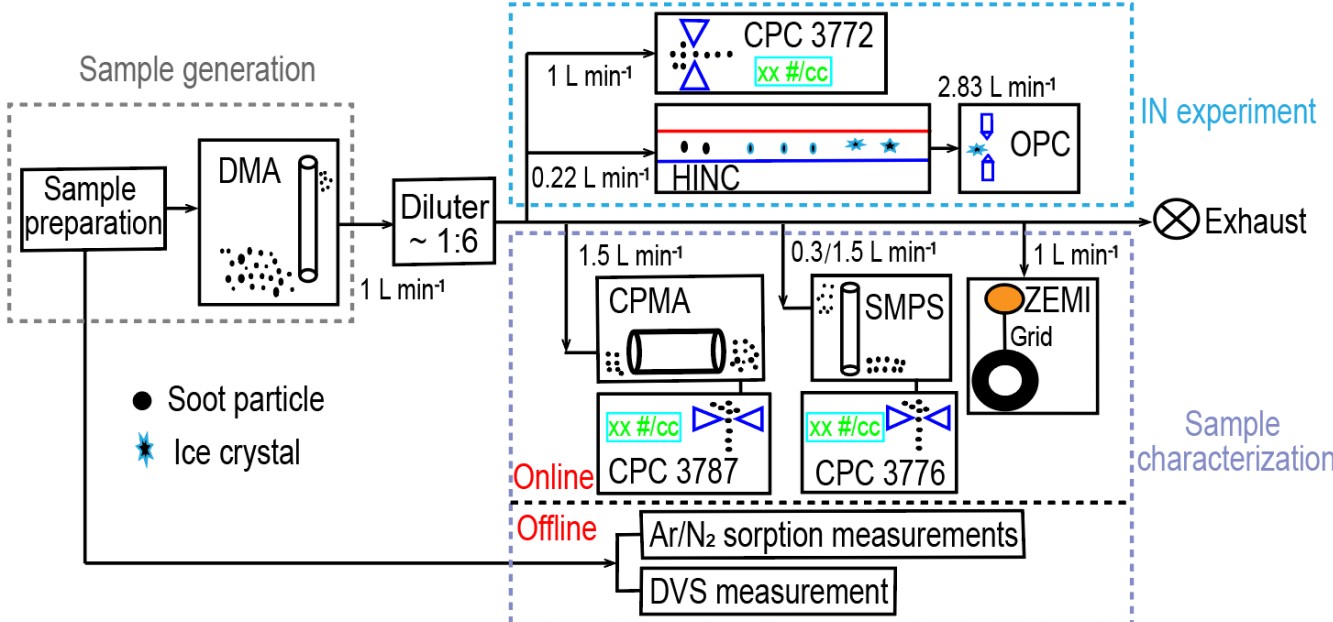

**Figure 1. The experimental schematic. The arrows show aerosol flow direction. DMA-Differential Mobility Analyser; CPC-Condensation Particle Counter; HINC-Horizontal Ice Nucleation Chamber; OPC-Optical Particle Counter; CPMA-Centrifugal Particle Mass Analyser; ZEMI-Zurich Electron Microscope Impactor; DVS-Dynamic Vapor Sorption.**

## 2.1 Soot sample preparation and aerosolization

In this study, two different types of commercial carbon black were used as fresh soot samples, including FW200 and Printex90 (PR90, Orion Engineered Carbons GmbH, OEC, Frankfurt, Main, Germany). As shown in Fig. 2, the Teflon coated magnetic bar was used to stir and transfer a part of kinetic energy to soot powders causing the displacement of single primary particles, which directly changes the soot sample porosity and makes it more compacted and densified compared to the fresh sample. In the first several hours agitation, the mass of 400 nm FW200 (Fig. 3a) and PR90 (Fig. 3b) particles are ~ 10.8 and ~ 13.2 fg initially and the mass value increases to a plateau value ~14.3 and ~ 16.2 fg after a 3-hour agitation. The particle mass of a fixed mobility size continues to increase, which means the particle effective density is also increasing, signifying compaction. As shown in Fig.3, size selected 400 nm soot particle mass is reasonably constant after two-weeks agitation as the mass variability falls within the uncertainty range of ± 0.6 % and ± 1.5 % for FW200 and PR90 soot 400 nm particles, respectively. This indicates the maximum degree of compaction is achieved within 3 hours of mechanical agitation. When the same sample has been stirred/compacted for additional two weeks, no further mass increase was detected. Compared to the fresh soot sample, the number size distribution mode of the polydisperse agitated particles shifts to a smaller value (Fig. A1), showing that the peak size mode shifts from ~ 257 nm for fresh FW200 soot to ~ 219 nm for the densified sample, and that of fresh PR90 soot changes from ~ 300 to ~ 264 nm. The change in aerosol particle size distribution (Fig. A1) covers the size range from ~ 50 to 400 nm and ~ 60 to 800 nm for FW200 and PR90 soot, respectively, which encompasses the monodisperse size





ranges investigated here and suggests soot particle morphology change induced by agitation occurs for the size range investigated in this work (60 to 400 nm).

In the following experiments, fresh soot powder was dispersed into soot aerosol and referred to fresh samples, termed
FW200fresh and PR90fresh. Compacted soot particles, called FW200comp and PR90comp hereafter, means soot samples which were agitated for at least two weeks and that the mass of size selected soot particle already reaches the stable level similar to the case shown in Fig. 3. In total, four kinds of soot samples (FW200, FW200comp, PR90 and PR90comp) were used in this study at 4 different sizes. Simultaneously with agitation, the dry soot powder while being stirred (fresh or compacted for more than two weeks) was aerosolized with a $N_2$ flow of velocity of $\sim 35$ m s$^{-1}$ through a venturi nozzle creating
large enough shear forces to break down dispersed soot agglomerate into small aggregates. A distinction of this soot particle 'aging' method is that no chemical aging effects are involved, thus allowing exclusively to investigate the influence of morphology changes on soot IN ability.

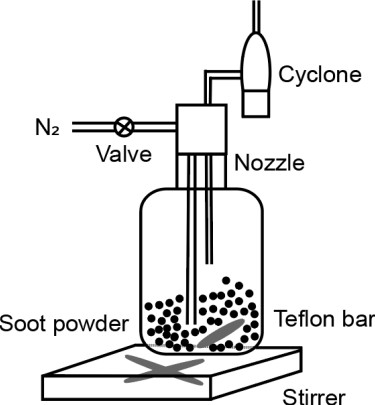

**Figure 2. The schematic of soot aerosol sample generation setup.**

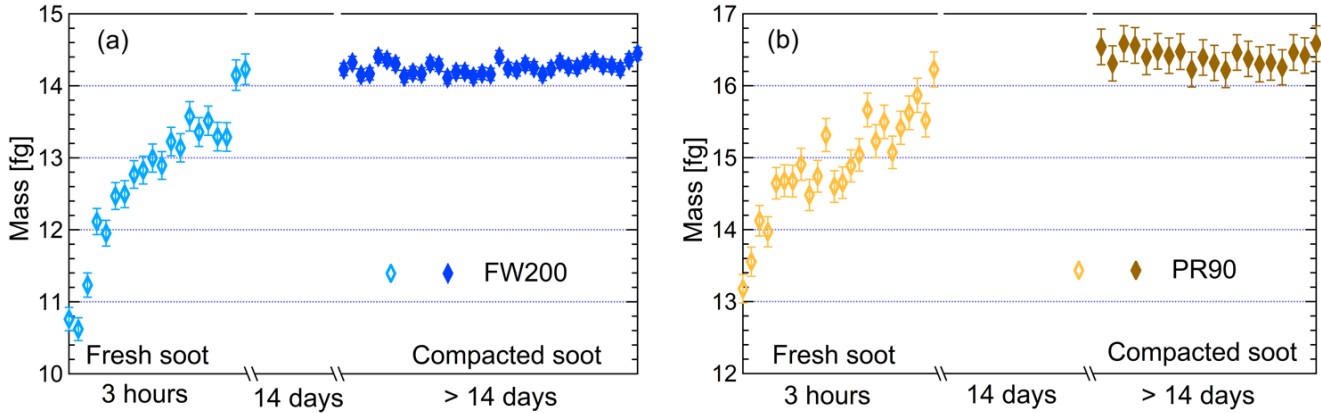



**Figure 3. The mass of 400 nm FW200 (a) and PR90 (b) soot particles as a function of agitation time. The measurement uncertainty mainly originated from CPC counting (± 10%) (Olfert et al., 2006). The error bars for fresh samples indicate the upper limit of the uncertainty and the error bars for compacted samples represent one standard deviation of corresponding measurements.**

### 2.2 Ice nucleation experiments

The IN experiments were performed by running RH scans in HINC from ice saturation ($S_i$) values of unity to water saturation ($S_w$) conditions of 1.1 at $T$ ranging from 243 to 218 K with a step of 5 K. The HINC RH scan was conducted at a $S_i$ rate of $0.02 \cdot min^{-1}$ at each $T$. The principle of HINC has been explained in detail by Lacher et al. (2017) and Mahrt et al. (2018). In brief, HINC is a continuous flow diffusion chamber and creates supersaturation conditions with respect to ice and/or water by using the difference between the diffusion rate of water vapor and air. Two copper plates coated with an ice layer at different

$T$ ($T_{top} > T_{bottom}$) form the nucleation and growth parts of the chamber. The aerosol sample flow is jacketed between two $N_2$ sheath flow with a ratio of 1:12 and is confined in the centre of the chamber, where the $T$ and RH conditions for the aerosol particles can be controlled by adjusting the $T$ difference between $T_{top}$ and $T_{bottom}$. The RH uncertainty range for each RH and $T$ condition during the RH scan in the chamber is influenced by the $T$ uncertainty of four thermocouples mounted on each chamber wall, which is ± 0.1 K. According to Mahrt et al. (2018), the RH uncertainty at $T = 218$ K is reported to be ± 5% $RH_w$

when conditioning the chamber at 105 % $RH_w$ with a sample to sheath flow ratio of 1:12, and the uncertainty is smaller at warmer $T$ and lower RH conditions. By feeding in soot sample aerosol with a movable injector into the chamber at a fixed position, the IN activity of the soot particles with a transition time ~ 14 s at a fixed $T$ and RH condition was measured. In order to calculate the activation fraction (AF, see Eq. (1)) ratios of the soot particles, a condensation particle counter (CPC; Model 3772, TSI Inc.) was used to monitor the number concentration of particles entering the chamber, and the number of water

droplets or ice crystals exiting the chamber is recorded by an optical particle counter (OPC; MetOne, GT-526S) in six size bins, including 0.3, 1.0, 2.0, 3.0, 4.0 and 5.0 μm. The AF ratio is defined as:

$$AF = \frac{n_{>1 \, \mu m}}{n_{total}} \quad (1)$$

where $n_{>1 \, \mu m}$ is the number of particles (activated as water droplets or ice crystals) larger than 1 μm recorded by the OPC and $n_{total}$ presents the total number of soot particles entering the chamber as recorded by the CPC.

### 2.3 Particle characterization methods

To characterize the morphology and porosity of soot particles in this study, both online soot aerosol aggregate and offline bulk soot sample powder measurements were performed. These measurements aim to investigate the changes in the compacted soot samples compared to the fresh samples for soot aggregate mass and size, $D_f$, microscopic TEM images and soot-water interaction ability, as well as soot sample PSD, $N_2$-BET (Brunauer–Emmett–Teller) specific surface area and porosity. The

relevant properties are summarized in Table 1.



### 2.3.1 Online soot aggregate characterization

Online soot aggregate property measurements mean that the soot particles for characterization were taken from the same aerosol flow as for IN experiments. Thus, the characterized soot particle population is identical to ice or water activated soot INPs or CCN. Considering that a HINC RH scan at a fixed $T$ takes ~ 40 min, during which the fresh soot might become

compacted to some extent (see Fig.3), soot samples were always replaced with fresh soot for every sample RH ramp during the IN experiments. Soot samples agitated for two-weeks were used to produce compacted soot aerosols. Both fresh and compacted 60, 100, 200 and 400 nm soot particles were sampled. In order to monitor the changes in the particle morphology and its influence on soot IN ability, the particle size and mass distribution of size selected soot was measured synchronously with IN experiments. The single particle mass of size selected soot aerosol sample was derived from the CPMA mass scan

measurements running with a CPC (Model 3787, TSI Inc., 1.5 L min$^{-1}$). Simultaneously, the size distribution of monodisperse soot particles was scanned by the SMPS setup operating with a CPC (Model 3776, TSI Inc.) in high flow mode (1.5 L min$^{-1}$) for 60 and 100 nm or in low flow mode (0.3 L min$^{-1}$) for 200 and 400 nm soot particles. Both the CPMA and SMPS raw data scans were fitted by a bimodal lognormal distribution model. The uncertainty in particle mass and mobility diameter was derived from the standard deviation of the mode mass from the CPMA scans and mode mobility diameter from SMPS scans

from at least 20-100 runs per sample. The mass-mobility exponent $D_f$ was obtained by plotting the single particle mass as function of the particle mobility diameter and by fitting the data with a power function. The Eq. (A1) in Appendix A is used to fit mass-mobility data. In addition, the size distribution of polydisperse fresh and compacted soot particle was also measured and presented in Appendix A. The proportion of double-charged particles is less than ~ 25 % and ~ 29 % for the 60 and 100 nm size selected particles, respectively, and approximately smaller than 20 % for the 200 and 400 nm (see Figs. A2 to A5). In

order to take TEM images for soot aggregate morphology analysis, soot aggregates were deposited on 400 mesh Cu girds with a formvar/carbon support film (TED, PELLA, INC.), utilizing the ZEMI setup (Aerni et al., 2018; Mahrt et al., 2018). The sample grids were visualized at a magnification value of 75k.

### 2.3.2 Offline soot particle characterization

Bulk soot sample property measurements, termed as offline measurements in Fig. 1 including Ar, N$_2$ and water vapor gas

physisorption isotherm measurements, were not conducted simultaneously with IN experiments. Soot powder samples for these offline measurements were prepared in the same way as for online measurements. Therefore, different standard methods and analysis approaches are used to characterize soot cavity characteristics. In the following, we describe the different gas physisorption measurements used and associated analysis models applied.

**Gas physisorption measurements: N$_2$, Ar and water vapor**

Gas physisorption measurements can be used to evaluate particle structural and surface properties by measuring the interaction characteristics of bulk particulate sample with respect to probe gas pressure levels. Different probe gases have their advantages



and disadvantages to address specific particle properties. $N_2$ is widely used for physisorption measurements to characterize porous material PSD in the literature (Jelinek and Kovats, 1994; Kruk et al., 1997; Galarneau et al., 1999; Hayati-Ashtiani,

2011; Kupgan et al., 2017). Some studies reported that the results from $N_2$ physisorption measurements might be biased due to its quadrupolar nature and the interaction with the substance surface functional groups, which affects the orientation of the molecule adsorbed. For instance, pore structure diameters for porous material derived from $N_2$ isotherms can be substantially underestimated (Jelinek and Kovats, 1994; Kruk et al., 1997; Lowell et al., 2004). More recently, Ar has become a more favoured noble gas in some laboratory studies (Gardner et al., 2001; Thommes et al., 2012; Sing, 2014b) as a result of its single

atomic shape and its inertness with the substrate surface polarities. Thus, we performed both $N_2$ and Ar isotherm measurements to compare the differences of soot cavity property results derived from both measurements. The DVS measurement, which has the advantage to reflect the interaction ability of soot with water vapor, was also conducted to understand soot IN activities at low $T$ with supersaturated water vapor. However, quantitative water vapor isotherm data analysis require the water-soot contact angle, which is not well constrained in the literature. Thus, contact angle assumptions are made for approximate calculations

in this study. By comparing the results of these three measurements, morphology differences between fresh and compacted soot particles and the influence on soot particle IN activities is discussed.

The Ar and $N_2$ adsorption and desorption isotherms for soot samples were measured at 77 and 87 K respectively at varying adsorbate gas pressure. Before the measurement, all samples were outgassed for 2 hours at 573 K under vacuum. The raw data

is presented as the volume of the adsorptive gas adsorbed by per gram of sample at standard temperature and pressure (STP) corresponding to each relative pressure ($p/p_0$) condition, which is indicated by the ratio of adsorptive vapor pressure ($p$) to the adsorptive vapor saturation pressure ($p_0$).

Before the DVS measurement, soot samples were dried for 4 hours at 573 K and then cooled down to ambient $T$ under vacuum.

The measurement was conducted at 298 K by a gravimetric dynamic method measuring the mass of water vapor adsorbed or desorbed by the soot sample at varying $p/p_0$ (or RH) conditions, assuming a quasi-equilibrium state of the soot sample under this vapor pressure. The initial sample mass is used as a reference for following mass measurements. For each sample, an adsorption isotherm was obtained from $p/p_0$ = 5 % to 95 % with a step of 5 % and then a desorption isotherm down to $p/p_0$ = 5 % with the same $p/p_0$ resolution was also measured. Finally, the sample mass change ($\Delta m$) in percentage as a function of

$p/p_0$ can be obtained, representing the amount of water adsorbed or released by the per gram soot sample at various $p/p_0$ conditions.

**Pore size distribution (PSD) analysis**

Sorption isotherms can be used to calculate the PSD which is of interest to describe the structure of carbon black aggregates.

In the following we apply the term "pore" to describe the cavity structures in the soot aggregate. The corresponding term 'pore diameter' is hereby defined as a diameter equivalent to the cylindrical pore width that leads to the same vapour pressure


reduction due to the inverse Kelvin effect. The PSD results reflecting soot cavity characteristics as a function of pore size can be derived from the desorption isotherms by applying the Kelvin equation with specific assumptions. The Kelvin equation is given as the following:


$$r_k = -\frac{2\gamma_{sl}v_s\cos(\theta)}{RT ln\left(\frac{p}{p_0}\right)} \quad (2)$$

Where $r_k$ is the pore radius required for capillary condensation induced by inverse Kelvin effect, $\gamma_{sl}$ denotes the interfacial tension between the solid and the liquid phase, $v_s$ is the adsorptive molar volume, $\theta$ is the contact angle between soot surface and the adsorptive gas which can be taken as 0° for $N_2$ and Ar, $R$ is the ideal gas constant with the value of 8.314 J mol$^{-1}$ K$^{-1}$. The application of the Kelvin equation for PSD analysis relies on the following assumptions: (1) the equation is valid over the

complete pore size range addressed; (2) the pores are rigid and of a cylindrical structure; (3) the capillary filling or desorption of each pore does not depend on its location within the pore network inside the aggregate; (4) the adsorption or desorption on the pore walls proceeds exactly in the same way as on the corresponding open surface.

Liquid can condense in mesopores (2-50 nm) when $N_2$ or Ar saturation ratios are above the BET range ($p/p_0 > 0.15$ or 0.1),

due to capillary effects according to the Kelvin equation. This condensation step is reversible with decreasing $p/p_0$, which refers to the desorption branch of the isotherms and is in equilibrium state. By applying Brunauer-Joyner-Holanda (BJH) approach (Barrett et al., 1951; Lowell et al., 2004), the PSD can be calculated from the desorption branch of the respective sorption isotherm. A more detailed description for the PSD results determination by BJH approach can be found in International Union of Pure and Applied Chemistry (IUPAC) recommendation for calculation of the distribution of mesopores (Thommes

et al., 2015). The BJH approach calculates the pore size from the Kelvin equation at each $p/p_0$ value and the cumulative adsorbed gas volume corresponding to a pore size value can also be retrieved. However, the desorption step includes the desorption from the adsorbent surface in addition to an release of $N_2$ or Ar from pores and these surface desorbates are irrelevant to pore structures during the desroption process. To distinguish between the liquid released from pores and that which desorbed from the material surface, the De-Beor thickness equation was applied (Thommes and Cychosz, 2014;

Thommes et al., 2015). With the BJH method, pore diameters from 2 to 400 nm can be calculated, but values close to the limits have to be interpreted carefully. At the lower limit, inter-molecular forces lead to an enhanced condensation of liquid. Therefore, the pore diameter below 10 nm calculated by the Kelvin equation is an underestimation of the real pore diameter by approximately 30 % (Lowell et al., 2004). The upper limit of 400 nm is defined by the highest $p/p_0 \approx 0.99$ that can be reached during the measurement. Due to the irregular geometry of carbon black aggregates, the intra-aggregate cavity volume among

primary particles cannot be clearly distinguished from the outer-aggregate volume between primary particle clusters. Therefore, the pore volume at the highest pore diameter is uncertain. Regardless of the limitation of BJH method, the calculated PSD results can be used to compare different types of soot and to investigate the effect of particle compaction on the spatial pore density of soot aggregates in this study.





Analogous to the BJH method applied for $N_2$ and Ar isotherms, a PSD analysis approach can be formulated for DVS data on the basis of the Kelvin equation as well. The DVS isotherm PSD analysis method is based on the theory proposed by Wheeler (1955) and more recently formulated by Shkolnikov and Sidorova (2007), and is also similar to the approach applied by Mahrt et al. (2020b). A six or seven order polynomial function is used for the extrapolation of the discrete isotherm data points into a polynomial expression instead of a three order polynomial stepwise fitting for every three data points used by Mahrt et al.

(2020b). The desorption branch in water vapor sorption measurement was used to derive the PSD as this branch is more associated with the equilibrium gas-liquid phase transition. Because soot-water contact angle was not measured, three contact angle values including 0°, 45° and 75°, representing surface wettability from hydrophilic to hydrophobic, were used to cover the possible values for soot samples in this study. With a contact angle value assumption, pore radius can be calculated as a function of $p/p_0$, according to Eq. (2). At each water vapor $p/p_0$ condition, the mass of water vapor is measured in the DVS

measurement. Subtracting the amount of water sticking on the material surfaces and wall area, water volume can be calculated which is equivalent to the pore volume. Hence, the pore volume distribution can be determined as a function of pore radius. A detailed formulation of this approach is provided in Appendix B.

**Table 1. Soot sample characterization results.**

| Soot type | FW200 Fresh | FW200 Comp | PR90 Fresh | PR90 Comp |
|---|---|---|---|---|
| [@]Volatile matter at 950 ℃ (wt/wt%) | 20 | | 1 | |
| [@]Primary particle size nm | 13 | | 14 | |
| [#]Fractal dimension ($D_f$) | 2.64 | 2.62 | 2.66 | 2.66 |
| [*]$N_2$ $S_{BET}$ m$^2$ g$^{-1}$ | 552.4 ± 5.1 | 567.3 ± 7.3 | 336.5 ± 2.7 | 332.9 ± 3.0 |
| Total pore volume at $p/p_0 = 0.99$ ($N_2$) mm$^3$ g$^{-1}$ | 1543 | 1431 | 1135 | 755 |
| Total pore volume at $p/p_0 = 0.99$ (Ar) mm$^3$ g$^{-1}$ | 1788 | 865 | 967 | 662 |
| [&]the number of voids per aggregate | 1.88 (136) | 0.27 (147) | 1.71 (154) | 0.28 (143) |

[@]Information from the manufacturer

[#]Calculated from the SMPS-CPMA particle size and mass measurement results by using Eq. (A1)

[*]$N_2$ $S_{BET}$ is calculated by the formulation presented in Appendix B

[&]Defined as two-dimensional void encompassed by primary particle clusters per aggregate and derived from TEM image analysis. The total number of soot aggregates analysed is shown in the brackets, respectively.

**3 Results and discussion**

**3.1 Evidence for soot aggregate densification (compaction)**

**Particle effective density**: Soot particles are fractal aggregates consisting of numerous spherical primary particles. Primary particles form branch-like structures, which are not rigid and susceptible to rearrangement by mechanical forces upon physical





agitation. The particle effective density ($\rho_{eff}$) can be an indicator of the spatial density of intra-aggregate primary particles

within a given mobility size soot aggregate. As shown in Fig. 3, the mass change of 400 nm size selected soot particles after

physical agitation shows a $\rho_{eff}$ increment. The $\rho_{eff}$ is calculated by the following equation:

$$\rho_{eff} = \frac{6m}{\pi \cdot D_m^3} \quad (3)$$

where $m$ is the particle mass, which is an average value from the CPMA measurements, and $D_m$ is average particle mobility

size derived from the SMPS measurements. The $\rho_{eff}$ results at all sizes for both fresh and compacted particles are presented

in Fig. 4, showing the $\rho_{eff}$ of size selected soot particles for the IN experiments at different $T$. The agitation process increases

FW200 soot particle $\rho_{eff}$ by more than 10 % and for PR90 soot by more than 15 %, showing that all compacted FW200 and

PR90 particles are denser than the fresh with the same mobility size. This evidence on soot aggregate compaction, suggests its

cavity structure and intra-aggregate void volume (pore size) could be modified as a result of the mechanical stirring and

potentially resulting in mesopore enrichment.

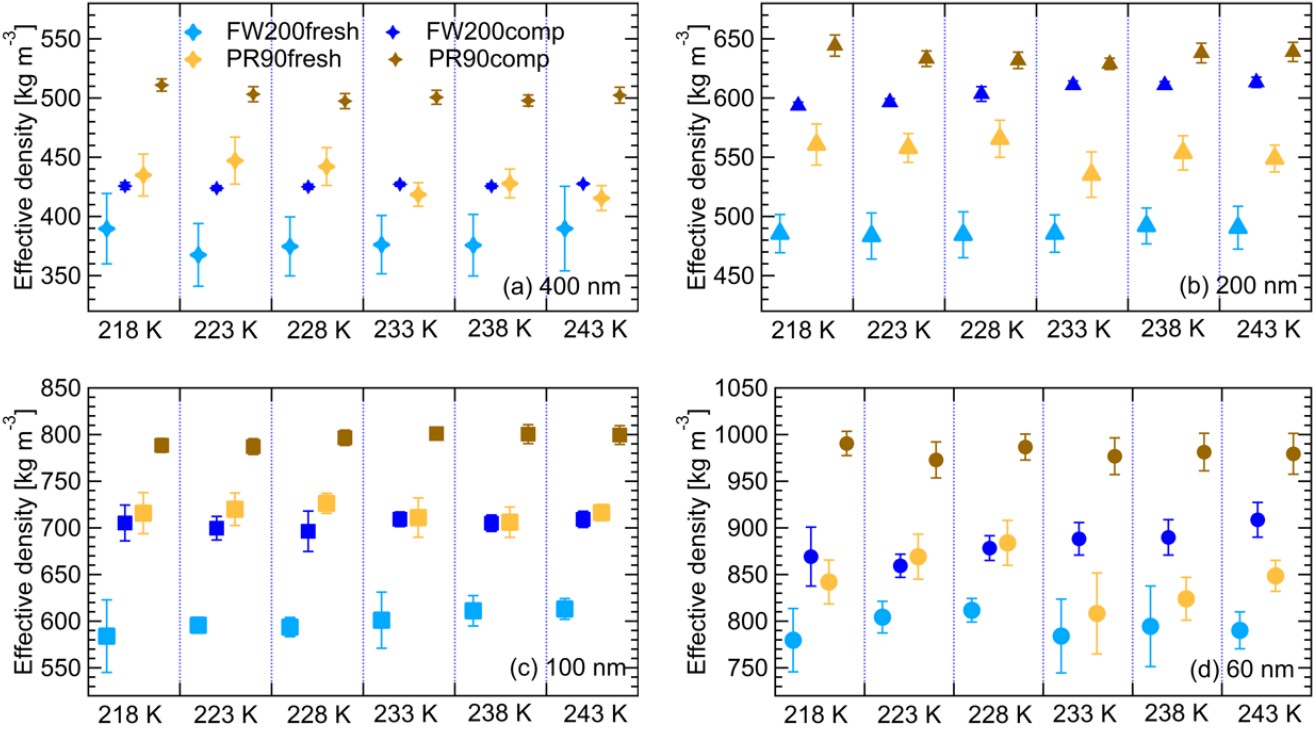

**Figure 4. The effective density ($\rho_{eff}$) of 400 nm (a), 200 nm (b), 100 nm (c) and 60 nm (d) size selected fresh and compacted FW200 and PR90 soot particles at each IN experiment $T$ (x-axis). The error bar is propagated from the particle mass and mobility diameter measurement uncertainties.**





**Soot aggregate compaction**: As shown in Fig. 5, visual microscopic evidence on soot structure compaction can also be confirmed. Intra-aggregate voids, indicated by red circles with size of tens of nano-meters, can be seen from fresh FW200 and PR90 soot aggregate images (Fig. 5a and c), showing a lacy or open-branched structure. Herein, the intra-aggregate void is defined as the two-dimensional cavity space encompassed by soot primary particle clusters in a soot aggregate. However, well agitated soot aggregate TEM images (Fig. 5b and d) show a compact and dense structure with a rare presence of cavity

structures. This finding coincides with higher $\rho_{eff}$ values for compacted soot shown in Fig. 4. For each type of soot sample, the number of voids per soot aggregate was calculated and provided in Table 1. In general, this kind of void can be observed for fresh soot aggregate whereas the void occurrence probability for compacted soot aggregates is much less than unity, demonstrating that the void structure universally exists in fresh soot aggregates and is reduced upon aggregate compaction. It can be inferred that the whole primary particle network or spatial pore structures inside a soot aggregate should have been

modified during the agitation process, generating compacted soot particles with a smaller PSD. These intra-aggregate voids may also explain why fresh soot samples have a larger total pore volume reported from gas sorption measurements (see Table 1). This finding agrees with the intra-aggregate void volume measurements results for bulk carbon black materials performed by Joyce et al. (2009), who suggested that the void volume decreases with increasing sample compression level.

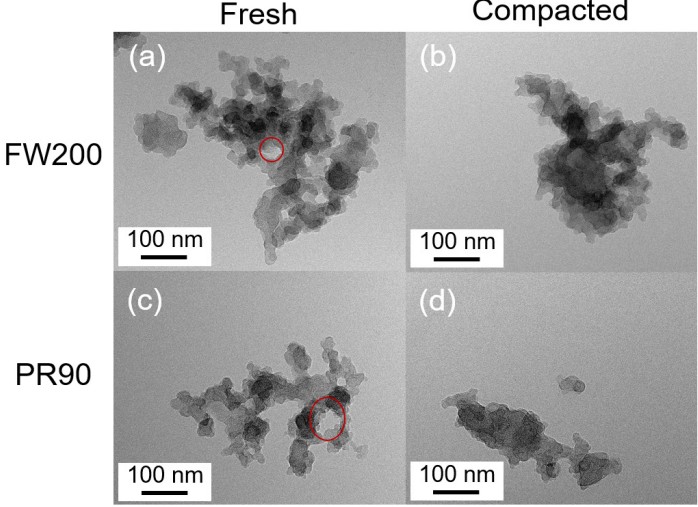

**Figure 5. TEM images for 400 nm fresh and compacted FW200 and PR90 soot aggregates at a magnification value 75k.**

### 3.2 Soot particle ice nucleation results

### 3.2.1 Fresh and compacted soot particle activation fraction curves

The AF curves for fresh and compacted FW200 and PR90 soot with different particle sizes are presented in Figs. 6 to 9,

respectively. Our soot particles only show water droplet formation at $T >$ HNT and nucleate ice at $T <$ HNT. Evidence on soot water droplet formation at $T >$ HNT is presented in Figs. C1 to C4, showing that OPC signals from the 5 μm channel are absent





for $T$ = 243 and 238 K. Since droplet growth rates are much lower than that of ice crystals due to the difference in $RH_i$ and $RH_w$, any ice crystals nucleating onto the soot particles at $T < 233$ K would be able to grow up to 5 μm in HINC and be detected by the OPC. This is also consistent to the results reported by Lacher et al (2017) and Mahrt et al. (2018). For $T <$ HNT, 200 and 400 nm compacted FW200 and 400 nm compacted PR90 soot particles generally can reach a higher AF value at the same $T$ and RH condition compared to the fresh particles, suggesting compaction enhanced IN activities at $T <$ HNT and soot-water interactions for $T >$ HNT. Aggregate compaction also promotes 100 nm FW200 soot activation at T ≥ 233 K as shown in Fig. 8. However, the rest AF curves for small size (60 and 100 nm) compacted and fresh FW200 or PR90 soot are within error bars. Overall, compacted soot particles with larger sizes are more effective INPs in cirrus regime compared to the uncompacted. We attribute the soot particle IN activity below $RH_{hom}$ at $T <$ HNT to the PCF process. In the following, we compare the differences in soot particle IN between fresh and compacted soot particles produced from both BC materials.

**FW200**: For 400, 200 and 100 nm compacted FW200 soot particles, our results are in good agreement with Mahrt et al. (2018), who studied the IN activity of the same BC material with the same sizes but utilized a fluidized bed aerosol generator (FBG) to produce soot aerosol samples. It is very likely the beads and rotating motion of the transition belt in the FBG setup can also result in soot aggregate compaction, which has a similar compaction effect on the fresh soot as induced by Teflon bar mechanical agitation in this study. However, fresh soot particles without physical ageing may have a loose structure. As shown in Fig. 6, 400 nm compacted FW200 soot particles are more effective INPs than the fresh sample as the compacted particles required a lower RH to reach the same AF value at $T < 233$ K. We ascribe this to an enrichment in small mesopores for compacted FW200 soot particles. After agitation, macropores (> 50 nm) and larger mesopores (> 20 nm) among fresh soot aggregate might be compacted into smaller mesopores which induce inverse Kelvin effect and lead to capillary condensation at RH conditions well below water saturation condition thereby promoting PCF activation, according to soot-PCF framework (Marcolli et al., 2021). Given that physical agitation is the only modification to soot samples, the enhancement of IN ability must result exclusively from the changes in morphology (resulting in mesopore enrichment) induced by soot aggregate compaction. At $T = 233$ K, 400 nm fresh and compacted FW200 soot particles show comparable AF values at $RH_w < 94$ % but FW200fresh AFs are lower than that of FW200comp at $RH_w > 94$ %, as shown in Fig. 6c. This is also because FW200comp soot contains more mesopores relevant to PCF activation compared to the fresh sample, showing that 400 nm FW200comp reaches higher AFs than the fresh particles. Because the 400 nm FW200fresh sample may reach its limit for PCF relevant mesopores whereas a larger fraction of FW200comp sample has more mesopores satisfying the PCF process when reaching at the same AF level at $RH_w = 94$ % for 233 K. At $T >$ HNT, the enriched mesopores by aggregate compaction can also promote water droplet formation for 400 nm soot particles, as shown in Fig. 6a and b, in which compacted soot can form more water droplets than the fresh at the same RH and $T$ condition. This promoted soot-water interaction ability also results from the pore structure enrichment induced by aggregate compaction, considering that mesopores can make contributions to soot water uptake by facilitating water capillary condensation (Persiantseva et al., 2004; Popovicheva et al., 2008b; 2008a). The IN activities of 200 nm FW200 soot particles are similar to those of 400 nm FW200 but smaller size soot samples (60 and100 nm)





only freeze homogeneously (see Figs. 7 to 9) at $T <$ HNT. At $T >$ HNT, these small size soot particles ($< 400$ nm) also form water droplets only above water saturation conditions. We believe the smaller sized aggregates (100 and 60 nm) do not possess enough mesopores of the right size or structure due to the limited number of primary particles making up these smaller aggregates resulting in limited intra-aggregate void volume.


**PR90:** Size selected 400 nm PR90 soot particles show similar but less pronounced IN enhancement after compaction, in comparison to FW200 soot. At $T \leq 228$ K below HNT, PR90comp soot particles reach a higher AF value at the same $T$ and RH condition compared to PR90fresh, which means PR90comp are more active INPs. This is also because compacted PR90 soot contains mesopores of relevant properties (size and structure) to promote PCF activation. However, there is no difference

between the PR90comp and PR90fresh soot at $T = 233$ K (see Fig. 6c). This is because of the homogeneous freezing dependence on $T$ and the limitation of ice embryo growth with a small mesopore volume in PR90 soot. In addition, the homogeneous freezing nucleation rate constant decreases exponentially with increasing $T$ (Ickes et al., 2015). Additionally, the small PSD (see Sect. 3.4) in PR90 soot aggregates limits the volume of supercooled water engaging in pore water homogeneous freezing. Both inhibits PCF process by leading to a small homogeneous freezing rate (Ickes et al., 2015; Vali et

al., 2015; Koop and Murray, 2016). On the other hand, the pore volume in PR90 soot aggregate may be too small for the volume of water to freeze on the time scale of our experiments ($\sim 10$ s) or to allow the ice embryo to grow outside of the pore. David et al. (2020) reported a similar case that silica particles with smaller mesopores (2.8 or 3.3 nm) are less active INPs via PCF than those particles with larger mesopores (9.1 nm) at 233 K. At $T >$ HNT, PR90comp 400 nm soot particles do not tend to be more effective CCN than PR90fresh suggesting the enrichment of small mesopores caused by agitation has limited

influence on promoting PR90 soot-water interaction abilities. Smaller size ($< 400$ nm) PR90 soot particles do not show significant difference for IN ability between aggregates with and without compaction and require homogeneous condition to form ice crystals (Figs. 7 to 9) at $T <$ HNT. At $T >$ HNT, small size ($< 400$ nm) PR90 soot particles form water droplets at $RH_w > 105$ %, which implies their hydrophobic surface and low water interaction ability.

There are also differences between FW200 and PR90 IN activities. Firstly, PR90 soot particles are less active INPs compared to FW200 as shown that the same size fresh or compacted PR90 soot particles require a higher RH than that of FW200 soot with the same compaction level to reach the same AF value at the same $T$. According to the PCF mechanism (Marcolli, 2014; Marcolli et al., 2021), this probably results from the differences in soot sample properties, including sample PSD and water-soot contact angle, both of which are determinators for the PCF mechanism. Thus, PR90 soot PCF process occurs at higher

saturation conditions and requires low $T$ if PR90 soot is originally less porous than FW200 and/or of a lower surface wettability. Secondly, aggregate compaction induced IN promotion for PR90 soot is not as significant as for FW200 soot. As can be seen from Fig.7, soot aggregate compaction can exert IN enhancement for 200 nm FW200 soot particles but not for PR90 soot particles. This is also determined by intrinsic nature of the soot sample and may suggest aggregate compaction induced mesopore enrichment for PR90 soot is not as strong as for FW200 soot or may simply be due to a higher contact angle of PR90.





At $T >$ HNT, where PR90 soot particles are poorer CCN and requires higher RH conditions to form droplets than FW200 soot with the same sizes, also suggesting a paucity of pore structures and a low water interaction ability of PR90 soot.

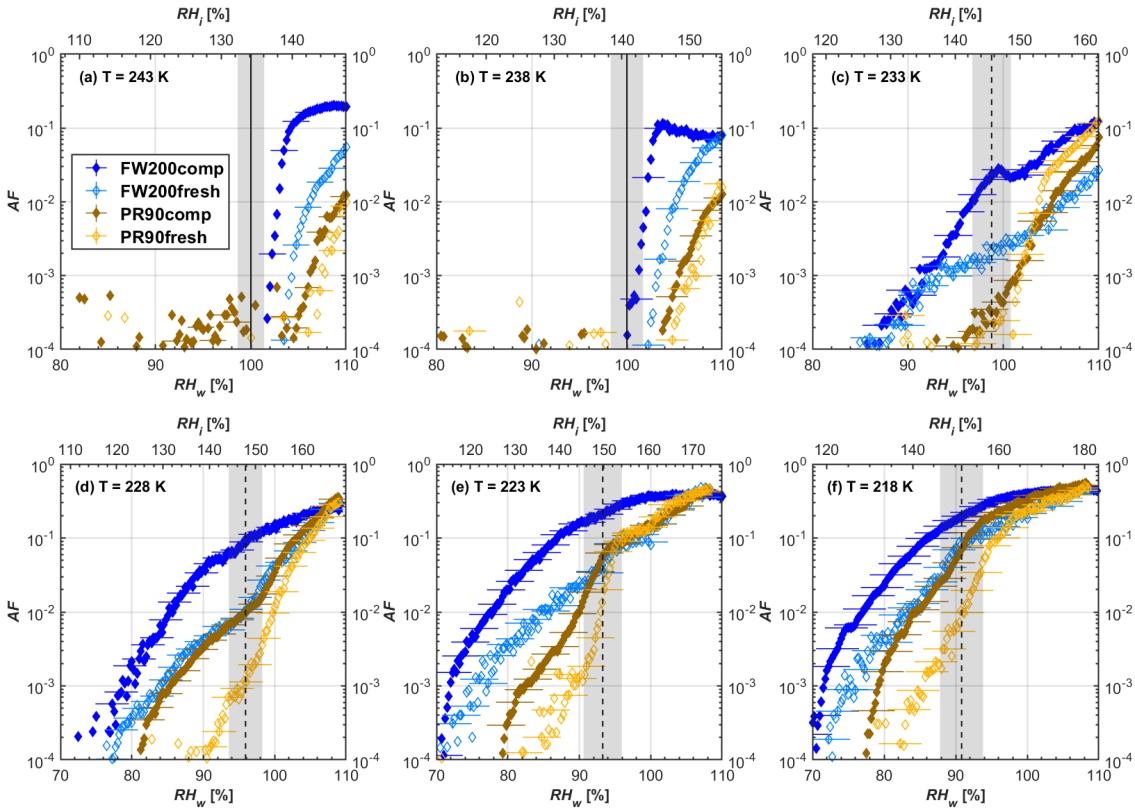

**Figure 6.** Average of three AF curves as a function of RH$_w$ and RH$_i$ from the 1 μm OPC channel for 400 nm fresh and compacted FW200 and PR90 soot particles at different $T$. For 243 and 238 K the AF curves represent water droplet formation (see text Sect. 425 3.2.1 for details). Black solid lines represent water saturation conditions according to Murphy and Koop (2005). Black dashed lines denotes the expected RH values for solution droplet homogeneous freezing at each $T$ (Koop et al., 2000). The grey shading shows the possible variation range in RH that aerosol in HINC can encounter for the calculated homogeneous freezing RH values at each $T$.

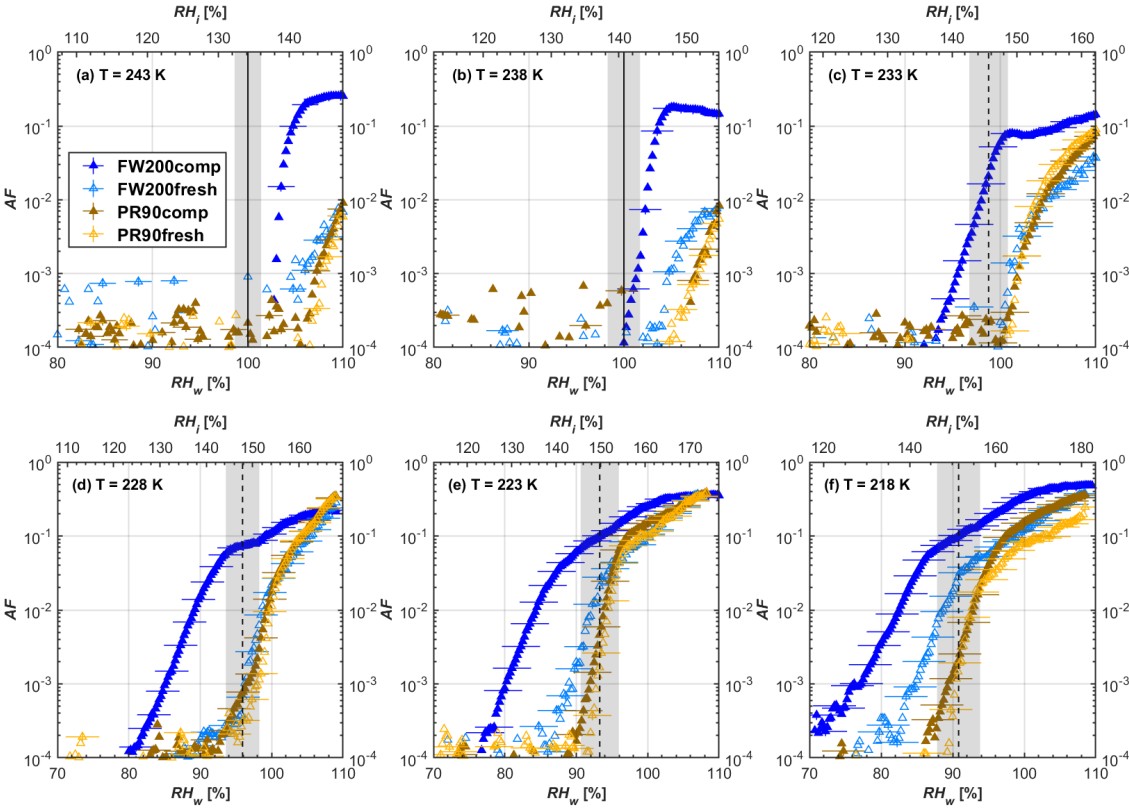

**Figure 7.** Average of three AF curves as a function of $RH_w$ and $RH_i$ from the 1 µm OPC channel for 200 nm fresh and compacted FW200 and PR90 soot particles at different $T$. For 243 and 238 K the AF curves represent water droplet formation (see text Sect. 3.2.1 for details). Black solid lines represent water saturation conditions according to Murphy and Koop (2005). Black dashed lines denotes the expected RH values for solution droplet homogeneous freezing at each $T$ (Koop et al., 2000). The grey shading shows the possible variation range in RH that aerosol in HINC can encounter for the calculated homogeneous freezing RH values at each $T$.





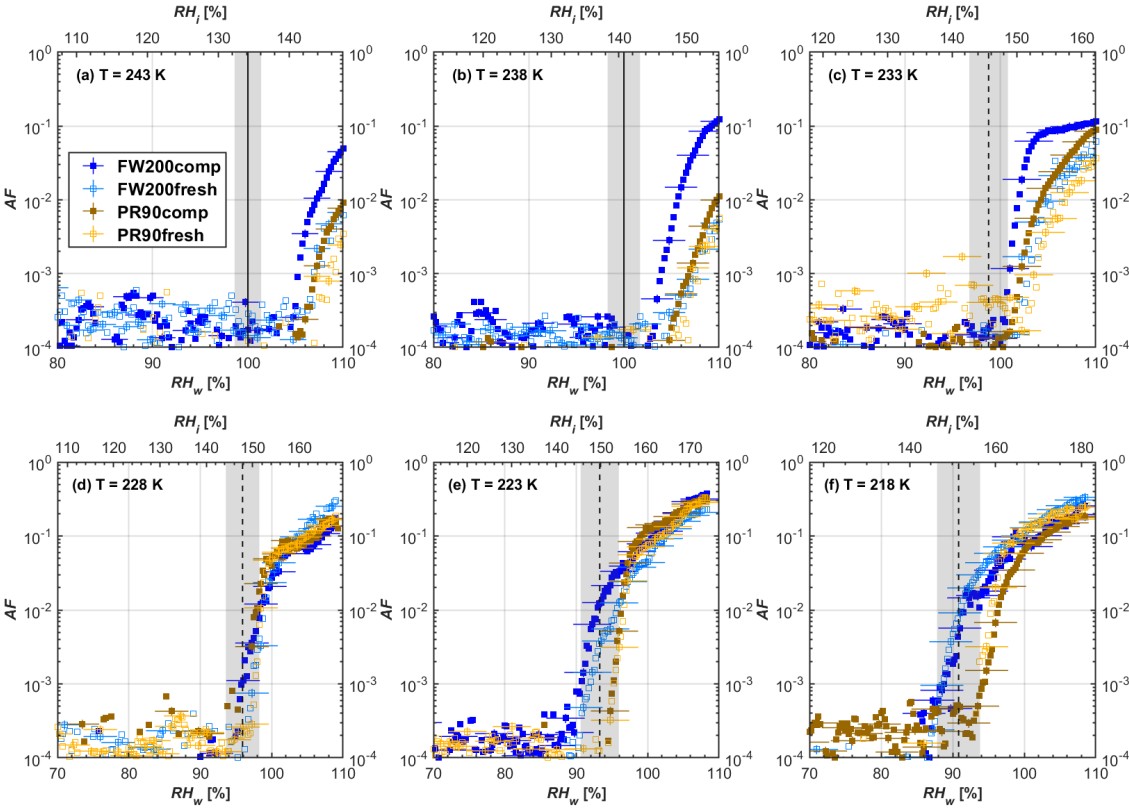

**Figure 8.** Average of three AF curves as a function of RH$_w$ and RH$_i$ from the 1 μm OPC channel for 100 nm fresh and compacted FW200 and PR90 soot particles at different $T$. For 243 and 238 K the AF curves represent water droplet formation (see text Sect. 3.2.1 for details). Black solid lines represent water saturation conditions according to Murphy and Koop (2005). Black dashed lines denotes the expected RH values for solution droplet homogeneous freezing at each $T$ (Koop et al., 2000). The grey shading shows the possible variation range in RH that aerosol in HINC can encounter for the calculated homogeneous freezing RH values at each $T$.



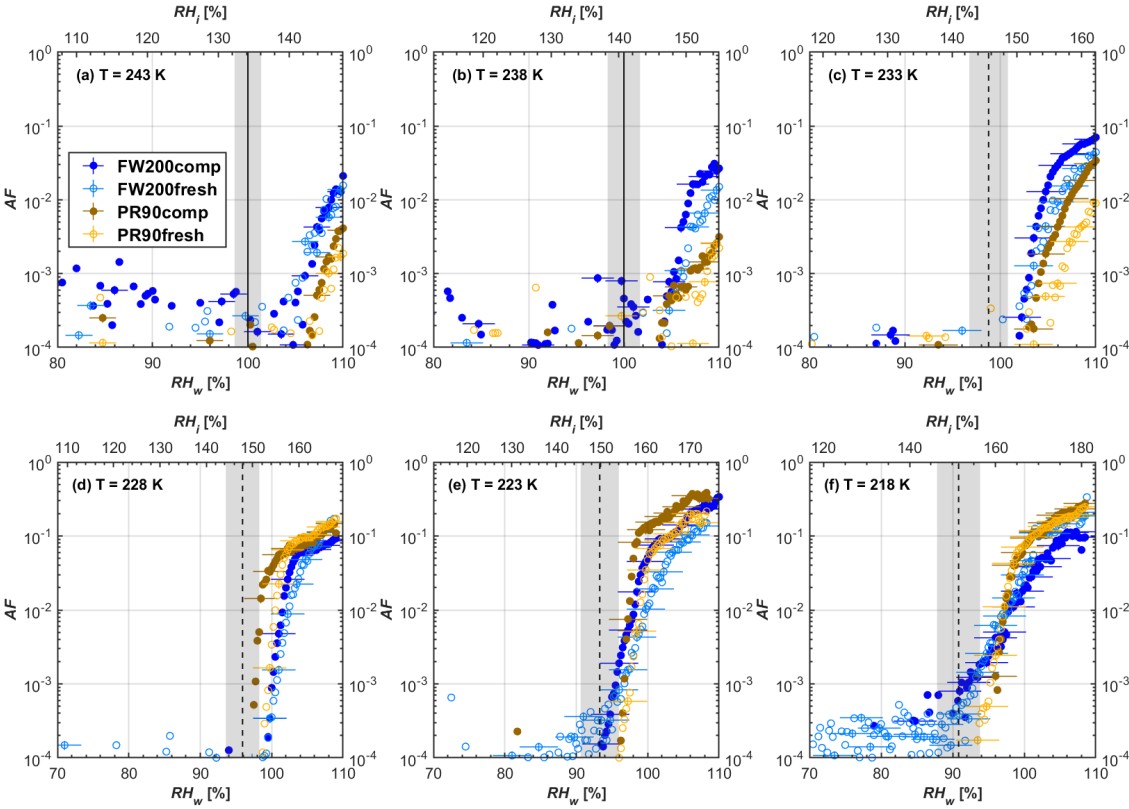

**Figure 9.** Average of three AF curves as a function of RH$_w$ and RH$_i$ from the 1 μm OPC channel for 60 nm fresh and compacted
FW200 and PR90 soot particles at different *T*. For 243 and 238 K the AF curves represent water droplet formation (see text Sect.
3.2.1 for details). Black solid lines represent water saturation conditions according to Murphy and Koop (2005). Black dashed lines
denotes the expected RH values for solution droplet homogeneous freezing at each *T* (Koop et al., 2000). The grey shading shows the
445 possible variation range in RH that aerosol in HINC can encounter for the calculated homogeneous freezing RH values at each *T*.

### 3.2.1 Fresh and compacted soot particle onset ice nucleation results

In this section, the IN ability of fresh and compacted FW200 and PR90 soot particles will be compared in terms of their onset
ice saturation ($S_i$) results, and the IN dependence on particle size will be discussed. The onset $S_i$ value is a measure to evaluate
the IN ability of INPs, using the threshold $S_i$ value required for the INP to reach a prescribed AF value (0.1 %) at a fixed *T*.
The onset $S_i$ values for each soot sample particle at different *T* are presented in Fig. 10, derived from the AF curves of the 1
μm OPC channel in Figs. 6 to 9 in which each curve is an average of three RH ramp experiments. In Figs. C5 to C8, we show
the size dependence for the sample type compared to Figs. 6 to 9 where we show different samples for a given size.



**FW200:** From Fig. 10a, soot aggregate compaction decreases the onset $S_i$ value for 400 nm FW200comp soot particles by at
least 0.08 compared to the fresh sample when $T < 233$ K in the cirrus regime. As shown in Fig. 10b, the onset $S_i$ for 200 nm
FW200 can be decreased by as much as 0.15 when $T < 233$ K. Particularly, there is also a reduction in onset $S_i$ value at $T =$
233 K for 200 nm FW200comp soot particles showing a smaller value just out of the error bar compared to 200 nm fresh
FW200 soot particles, which means the compaction induced IN enhancement still plays a role, albeit minor at $T = 233$ K. Both
compacted and fresh 100 nm FW200 soot particles are still competitive INPs than the solution droplets freezing
homogeneously at $T \leq 223$ K (see Fig. 10c). However, the onset condition values for fresh and compacted particles are
overlapped, which means limited IN enhancement for 100 nm compacted particles as discussed before. Finally, 60 nm FW200
soot particles cannot facilitate ice crystal formation below homogeneous conditions proposed by Koop et al. (2000) at $T <$
HNT, showing that the error bars are across the homogeneous freezing line even at the lowest $T = 218$ K. If we consider the
homogeneous freezing conditions of sulphate particles proposed by Schneider et al. (2021), 60 and 100 nm soot particles are
active INPs via PCF at $T \leq 223$ K as their $S_i$ values are below the corresponding $RH_{hom}$ condition (blue dotted line, Fig. 10).
At $T >$ HNT all data points lie within error bars and above the 1 μm water droplet limit line (limit above which we cannot
distinguish between ice and water for solely based on the 1 μm OPC channel) for both compacted and fresh FW200 soot of all
sizes, suggesting compaction does not have pronounced effects on FW200 soot droplet activation activities. Kanji et al. (2020)
suggested that soot particles are poor INPs in immersion mode freezing at $T >$ HNT and can only form water droplets under
these conditions. These results are also consistent with the findings reported by Friedman et al. (2011), Mahrt et al. (2018) and
Falk et al. (2021) in which the authors noted that soot particles do not nucleate ice for $T >$ HNT.

**PR90:** In general, PR90 soot requires a higher onset $S_i$ value than FW200 soot with the same compaction level to reach the
same AF = 0.1 % at $T <$ HNT, indicative of less active IN abilities. Nonetheless, soot compaction significantly enhances 400
nm PR90 soot IN at $T \leq 228$ K similarly to that of FW200 soot discussed above. In Fig. 10a, 400 nm compacted PR90 soot
onset $S_i$ values can be reduced by more than 0.1 in contrast to the fresh particles. Fresh and compacted 200 nm PR90 soot still
can compete with homogeneous freezing at $T = 218$ and 223 K as the $S_i$ value is smaller than the $RH_{hom}$ condition but the error
bars intercept with the homogeneous freezing line calculated according to Koop et al. (2000) as depicted in Fig. 10b. PR90
soot particles with or without compaction of sizes 60 and 100 nm only freeze homogeneously at $T <$ HNT even considering
the higher limit of homogeneous freezing proposed by Schneider et al. (2021), suggesting that the agitation process does not
change the onset results as shown that onset value error bars are in touch at all $T$ (see Fig. 10c and d). At $T >$ HNT, all sizes of
fresh and compacted PR90 soot particles form water droplets at similar RH conditions at $S_w > 1.05$, suggesting soot particle
structure compaction does not influence PR90 soot water droplet formation as discussed previously.





**Figure 10.** Onset saturation values (at AF = 0.1 %) with respect to ice ($S_i$) as a function of $T$, soot particle size (indicated in each panel) for fresh and compacted FW200 and PR90 soot samples. Black dashed lines denote the homogeneous freezing conditions for solution droplets at $T <$ HNT, according to Koop et al. (2000). Blue dashed lines denote the homogeneous freezing conditions for solution droplets at $T <$ HNT, according to Schneider et al. (2021). Grey dotted lines represent constant water saturation conditions calculated based on Murphy and Koop (2005). Red dashed lines indicate the 1 µm size water droplet survival conditions for $T >$ HNT, where the phase of the particle > 1 µm can be either ice or liquid water. The $S_i$ uncertainty caused by the temperature uncertainty ($\pm$ 0.1 K) of the HINC chamber is indicated as error bars. Each data point was derived from the AF curves in Figs. C1 to C4 (see Appendix C) in which each AF curve is an average of at least three individual RH ramps.

In brief, aggregate compaction can enhance soot particle IN ability significantly by lowering the onset $S_i$ values required for PCF activation at $T <$ HNT in the cirrus regime and showing a clear size dependence. Firstly, soot aggregate compaction induced IN enhancement shows a size dependence, suggesting that this compaction effect on soot IN ability is more pronounced for large soot particles (> 200 nm) and that physical agitation might not compact the small soot aggregate morphology as effectively as for large soot aggregates to induce IN enhancement. The size dependence effect is observed for the IN of soot particles with the same compaction level, showing that large particles require a lower onset $S_i$ value than smaller



ones at the same *T*, which coincides with the results reported by Mahrt et al. (2018) and Nichman et al. (2019) and also what is generally known about mineral dust ice nucleation (Archuleta et al., 2005; Welti et al., 2009; Kanji and Abbatt, 2010). This is as a result of increased probability or abundance of pore structures in large soot aggregates containing right size mesopores to facilitate the PCF mechanism. Partly, this is because large soot aggregates contain a larger number of primary particles and are more susceptible to a primary particle network change caused by external forces, whereas small soot aggregates contain a

limited number of primary particles and are more analogous to the fresh soot basic unit which is much close to primary particle clusters or a single primary particle. Thus, large soot aggregates better present the aggregate compaction effect on ice nucleation. The PCF activation has a positive relation with the number of mesopores, implying that the PCF activation requires sufficient mesopores with the right characteristic for capillary condensation and ice nucleation within the mesopores. Furthermore, individual mesopores even with appropriate pore size for PCF probably plays a limited or implicit role in the

event. For instance, dynamic molecular simulation results suggests that a single 3 nm pore in silica slab does not facilitate ice formation whereas ice crystal can grow quickly in an array of pores with the same width under the same simulation environment (David et al., 2019), suggesting that a bridging effect from the pore network promotes the growth of pore ice to a macroscopic ice crystal. In this study, experiments for 60 nm FW200 and PR90 soot are performed to extend a lower size limit for the investigation of soot particle IN activities in the cirrus regime since sub-100 nm particles are often observed in

the upper troposphere (Petzold and SchrÖder, 1998; Twohy and Gandrud, 1998; Cziczo and Froyd, 2014). According to Zhang et al (2020), there might be a size threshold between 100 and 400 nm for soot particles to be active INPs under cirrus cloud conditions. However, it is difficult to define such a threshold size for soot particles to trigger PCF mechanism under cirrus conditions. As can be seen from aforementioned discussion, the IN ability dependence on size is different for FW200 and PR90 soot in this study, showing that PR90 is more dependent on its size than FW200 soot as 200 nm PR90 soot onset results

are close to homogeneous freezing line whereas these of 200 nm FW200 soot are well below the $RH_{hom}$ limit. Last but not least, our IN results for 60 and 100 nm FW200 soot particles suggest that freshly emitted aviation soot particles majority with a small mobility size ($\leq$ 100 nm) could act as INP candidates in the cirrus cloud regime if they have properties (pores and surface wettability) similar to FW200 soot and the homogeneous freezing limit considered is analogous to that proposed by the homogeneous freezing parameterization of Schneider et al. (2021).

**3.3 Soot-water interaction ability**

Water vapor sorption measurement is an approach to characterize the cavity properties and water interaction ability of porous material. In previous studies, Mahrt et al. (2020b) used DVS measurement to evaluate the PSD of $C_3H_8$ flame soot and David et al. (2020) utilized this measurement to estimate the water-solid contact angle of porous silica. According to the IUPAC (Sing et al., 1985; Thommes et al., 2015), sorption includes two aspects of interaction between adsorptive gas and THE solid,

i.e. adsorption and desorption. As a function of water vapor $p/p_0$ levels, the $\Delta m$ values of both adsorption and desorption branches for each sample are plotted in Fig. 11. Based on the IUPAC classification (Sing et al., 1985; Thommes et al., 2015), the measured isotherms for FW200 compacted and fresh soot can be classified as type IV isotherm and PR90 soot follows a





type V. Both types of isotherms indicate the presence of mesopores in the sample. Larger isotherm hysteresis area for fresh and compacted FW200 soot implies they contain more mesopores for capillary water condensation than PR90 soot.

Additionally, a larger isotherm slope for FW200 than PR90 soot with the same compaction level at $p/p_0$ < 0.3 indicates a more wettable surfaces and a lower contact angle for FW200 soot. Both lower contact angle and abundance in mesopore structures facilitate PCF process, generally supporting the discussion in Sect. 3.2. hence FW200 soot particles have more active water uptake ability than PR90 soot of the same compaction level are more effective INPs.

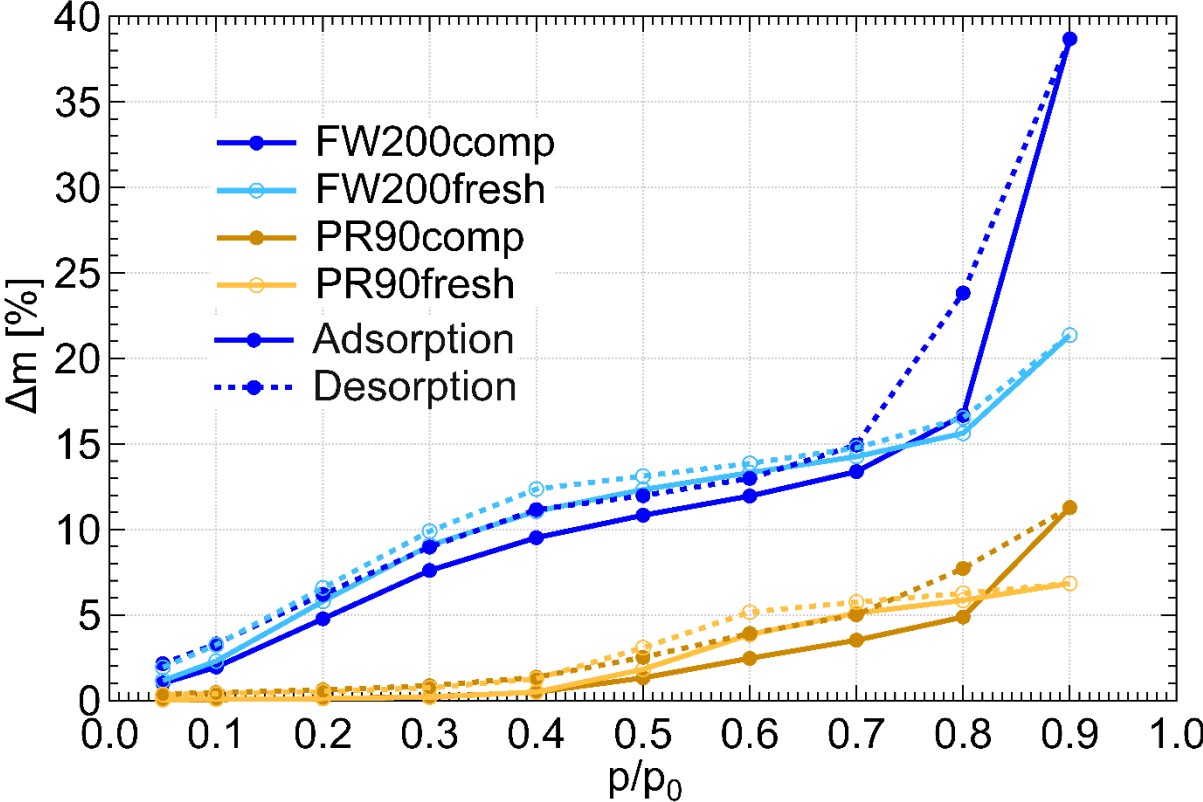

**Figure 11. Water dynamic vapor sorption (DVS) isotherms at 298 K for compacted and fresh FW200 and PR90 soot.** *Y*-axis indicates the mass change fraction ($\Delta m$) with respect to the original sample mass and *x*-axis stands for the relative water vapor pressure (*p/p$_0$*) levels. Solid lines indicate adsorption branches and the dotted lines indicate desorption branches.

**FW200 soot:** In general, soot-water interaction abilities are influenced by the density of local surface active sites, which are

hydrophilic and can accommodate water molecules more easily, and by soot porosity (Popovitcheva et al., 2000; Popovicheva et al., 2008b; Popovicheva et al., 2008a). At low $p/p_0$ range (< 0.3), soot water uptake is attributed to its active site density. Persiantseva et al. (2004) suggested that the water interaction of soot samples at low RH conditions is representative of its



water contact angle and a concave isotherm curve usually stands for a low contact angle. As shown in Fig. 11, isotherms for fresh and compacted FW200 soot both show the same kind of sensitivity to increasing $p/p_0$, suggesting both soot samples
contain a considerable number density of active sites and have a similar water contact angle. Figure 11 also shows that there is an inflection point for FW200 soot around $p/p_0 = 0.4$ which means the completion of monolayer water coverage and the start of capillary condensation induced by fine pore structures. Furthermore, the higher water uptake ability for fresh FW200 around $p/p_0 = 0.4$ suggests it contains more micropores (< 2 nm) than compacted FW200 because small size pores can induce capillary condensation at lower RH than larger pores. In addition, vapor isotherm hysteresis implies the existence of mesopores. For
$p/p_0 > \sim 0.4$, water may form multiple layers over the soot surfaces and mesopore capillary condensation might be induced by inverse Kelvin effect which is indicated by the hysteresis loop when considering the desorption isotherm. The hysteresis loop for fresh and compacted FW200 soot can be attributed to the Type H4 (Thommes et al., 2015), which is typical for materials that are both micro- and mesoporous. A considerably larger hysteresis loop for FW200comp means the sample contains plenty of mesopores whereas a smaller loop for FW200fresh implies that there are a relatively lower number of mesopores. Assuming
the soot-water contact angle to be the same for both fresh and compacted FW200, a smaller relative pressure of hysteresis loop knee in desorption branch (near $p/p_0 = 0.7$) for FW200comp than that of FW200fresh (around $p/p_0 = 0.8$) indicates that the mesopore size distribution of FW200comp is also smaller given that small mesopores fulfil Kelvin equation at a low $p/p_0$ levels (Lowell et al., 2004). Note that the hysteresis in desorption branches for FW200 soot even at low $p/p_0$ levels suggest the presence of ink-bottle pore structures, which cannot release absorbed liquid with decreasing environmental RH (Lowell et al.,
2004). In brief, water sorption results demonstrate that agitation will enrich the mesopore structures in FW200 soot samples but should not change soot particle surface wettability. This supports the IN enhancement for compacted FW200 soot particles in PCF activation at $T <$ HNT.

**PR90 soot:** The Type V isotherm with Type H3 hysteresis loop (Thommes et al., 2015) for PR90 soot shows the existence of
smaller mesopores but a higher soot-water contact angle compared to FW200 soot. It is evident that both PR90 soot contains much fewer active sites for water molecule adsorption at low $p/p_0$ levels than those of FW200 soot as both PR90 soot present a lower isotherm slope and a smaller $\Delta m$ at low pressures as shown in Fig. 11, indicating that its contact angle is higher than FW200 (Kireeva et al., 2009). The flat isotherm at low pressures, suggesting a low ability for multiple layer water adsorption, also manifests the weak water interaction ability over PR90 soot sample surface, i.e. a hydrophobic surface (Lowell et al.,
2004). For $p/p_0 < 0.4$, there is no significant water uptake activity difference between PR90comp and PR90fresh soot, suggesting physical agitation cannot modify their surface wettability. In addition, fresh PR90 soot adsorption and desorption isotherms level over those of compacted PR90 when $p/p_0 > 0.4$ and reach the largest difference at $p/p_0 = 0.6$, then become smaller than those of PR90comp soot when $p/p_0 > 0.8$. This can be explained similarly to the case for FW200 soot and it suggests that compaction increases larger size mesopores but decreases small sizes mesopores or micropores in PR90 soot
which can already lead to capillary condensation at low $p/p_0$ levels. A larger hysteresis loop for PR90comp soot than PR90fresh for $p/p_0 > 0.4$ suggests that the agitation process increases the availability of mesopores responsible for capillary condensation.





Therefore, compacted PR90 soot contains more larger mesopores than the fresh but both PR90 soot samples are more hydrophobic with a higher water contact angle than FW200 soot. Thus, the water sorption isotherms for PR90 soot qualitatively explains the enhanced IN for PR90comp soot particles and the relatively poor IN ability compared to FW200 soot particles of
the same size and the same compaction level.

In conclusion, DVS measurements provide soot water interaction results consistent with soot particle IN results. Firstly, it reveals a lower water contact angle contributes to more active IN for FW200 soot than PR90 soot (see Sect. 3.2). Meanwhile, DVS results demonstrate that physical agitation does not change soot surface active site density for water uptake and the
contact angle. In addition, DVS isotherms demonstrates the porosity change in soot samples with and without compaction. Agitation induced aggregate compaction causes larger size mesopore enrichment and enhances water uptake ability supporting the IN ability enhancement. From 400 nm soot particle IN results in Sect. 3.2, aggregates compaction can make PR90 soot particles as active as fresh FW200, suggesting that change in soot particle morphology can make originally hydrophobic soot IN active just due to changes in mesopore structure, highlighting the very direct role of particle morphology on ice nucleation.

**3.4 Soot particle pore size distribution**

Soot aggregate pore structures need to be optimal to play a crucial role in the PCF process. On one hand, the pore needs to be small enough to induce inverse Kelvin effect and water capillary condensation while on the other, the pore size is required to be large enough to be super critical with respect to the ice embryo size in order to support its growth out of the pore (Marcolli, 2020; Marcolli et al., 2021). Given a defined pore structure, with an approximate contact angle value and at a known $T$, the
RH condition required for the pore capillary condensation and pore ice growth can be calculated, according to the soot-PCF framework developed by Marcolli et al. (2021). In general, there exists different pore types, including cracks, cavities or capillaries, which can be like wedges, cone, cylindrical and other geometric structures. For the sake of a PCF process, mesopores, which are defined as cylindrical with diameters of between 2 and 50 nm (Sing et al., 1985), are of the main interests in this study. The simplifications and assumptions for soot pore structure is introduced in Sect. 2.3.2 and will be held in this
section as well.

**3.4.1 Pore size distribution retrieved from Ar and N₂ isotherms**

According to the BJH approach, the PSD results for soot samples can be analysed from Ar and $N_2$ isotherms, which is independent of the contact soot-water angle. Based on Ar and $N_2$ physisorption measurement isotherms provided in Figs. D1 and D2, the calculated PSD results as a function of pore radius are presented in Fig. 12. The PSD results generally show that
physical agitation shifts soot mesopore size distribution to smaller size ranges in comparison to the fresh samples. Compacted FW200 soot PSD derived from Ar BJH changes to a markedly smaller distribution mode from ~ 27 nm (21 nm, $N_2$ BJH) to ~





7 nm (13 nm, $N_2$ BJH) compared to the fresh soot (see Fig. 12a). Analogous to FW200 soot, PR90 soot Ar PSD peak decreases from ~ 14 nm down to ~ 6 nm after compaction (see Fig. 12b). $N_2$ BJH analysis for compacted PR90 soot shows a PSD reduction from ~ 16 nm to ~ 7 nm compared to fresh PR90 soot. In general, shifting to small PSD for compacted soot samples

suggests that small mesopores with radii approximately < 10 nm can be appreciably enriched after agitation, which would favour the PCF process thereby enhancing soot IN via PCF. The mesopore enrichment contributes to compacted soot particle IN ability enhancement because the smaller the mesopore size is the lower RH will be required to trigger PCF mechanism at the same $T$. For example, Ar PSD result for FW200 soot, presenting a larger mesopore size distribution for pore radius from ~ 3 to ~ 12 nm, directly supports onset results in Fig. 10 as shown that 200 and 400 nm FW200comp soot shows ice nucleation

at much smaller RH to reach the onset AF value (0.1 %) at the same $T$ than the FW200fesh soot. According to PCF process (Marcolli, 2014), 400 nm FW200 soot particles with a 60° water contact angle assumption (Marcolli et al., 2021) need ~ 4 nm mesopores to present onset $S_i$ = 1.24 at 228 K (Fig. 10a) in order to induce capillary water condensation. If the contact angle is assumed to be 45°, the mesopore should be around 6 nm. From Fig. 12a, FW200comp soot contains more mesopores around 4-6 nm than the fresh explaining 400 nm FW200comp soot can show such a low onset $S_i$ value. Whereas 400 nm fresh FW200

soot containing less 4-6 nm mesopores requires higher onset $S_i$ at the same $T$. In general, the increased mesopore availability around 10 nm contributes to the enhanced IN ability of compacted soot particle because of the pore morphology changes upon aggregate structure compaction.

However, both fresh soot samples contain more large size pore structures than compacted soot by showing a cross in the PSD

curves of fresh and compacted soot sample derived from the same gas sorption technique. Fresh FW200 soot has more mesopores > ~ 12 nm (Ar PSD) or > ~ 18 nm ($N_2$ PSD) than the compacted soot sample and the similar case for PR90 soot is about > ~ 9 nm (Ar PSD) or > ~ 11 nm ($N_2$ PSD). These pore structures are too larger to trigger inverse Kelvin effects at the same RH condition as required by smaller mesopores (< 10 nm), making them PCF irrelevant. For example, a 12 nm mesopore in a 400 nm FW200 soot aggregate with a water contact assumption 60° requires $S_i$ = 1.38 at 233 K to induce capillary water

condensation according to the Kelvin equation, which is already higher than the onset $S_i$ value for 400 nm both fresh (1.36) and compacted (1.34) FW200 soot particles shown in Fig. 10a. Hence, these large mesopores in fresh soot particles does not contribute to the IN activity. Besides, fresh soot samples also contain more pore structures larger than 50 nm in comparison to the compacted soot (see Fig. 12), which is consistent with their larger cumulative pore volumes (Table 1). These larger pore structures are probably to be intra-aggregate voids (macro pores > 50 nm) among primary clusters or outer-aggregate voids as

shown in TEM images (Fig. 5), which can be filled by Ar and $N_2$ agent near the saturation condition during sorption measurements.





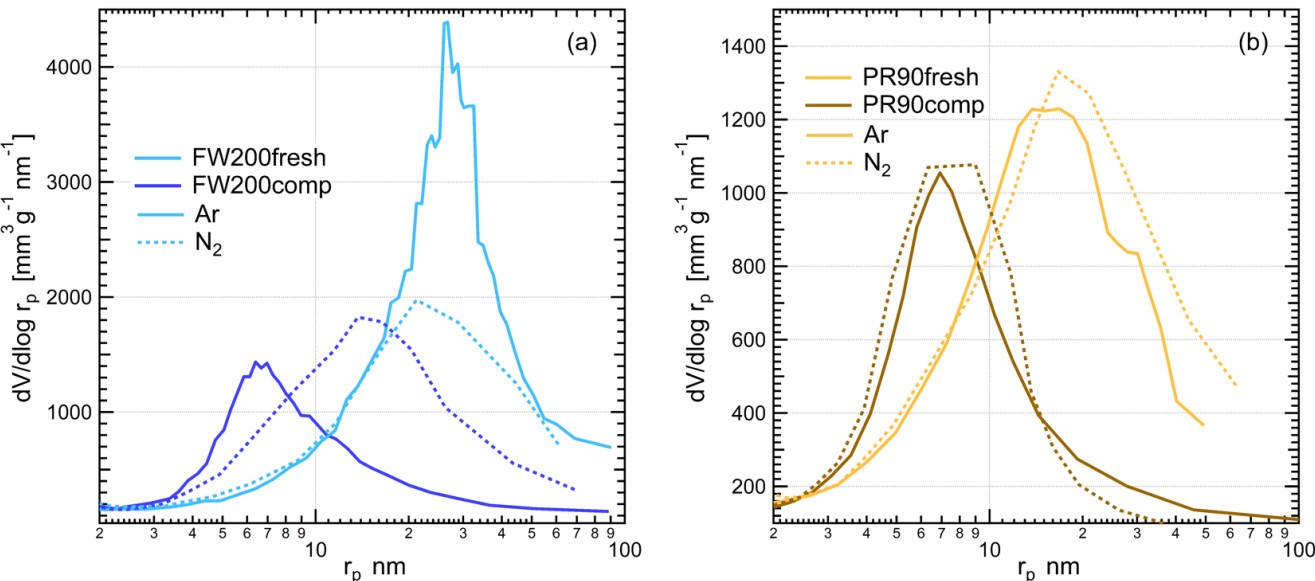

**Figure 12. The PSD as a function of pore size ($r_p$) for fresh and compacted (a) FW200 and (b) PR90 soot based on Ar and $N_2$ adsorption and desorption isotherms (Figs. C1 and C2). Solid lines and dashed lines are for results derived from Ar isotherms and $N_2$ isotherms, respectively.**

Gas sorption measurements also manifest the structure difference between soot samples and the derived PSD dependence on gas agent (Ar and $N_2$). Comparing the PSD results between FW200 and PR90 soot with the same physisorption measurement, there is a difference to be noted even though they reveal similar mesopore size change effects exerted by agitation. FW200 soot PSD curve can reach a much higher peak and contains a larger quantity of mesopores than PR90 soot with the same compaction level, supporting that FW200 soot is more active in PCF activation than PR90 soot (Sect. 3.2). A higher porosity level of FW200 soot than PR90 soot is also proved by its larger cumulative pore volume from BJH analysis (see Table 1). In addition, the agitation can lead to a more significant Ar PSD change to fresh FW200 soot than to fresh PR90 soot, showing that compacted FW200 soot Ar PSD peak mode reduction from ~ 27 to ~7 nm whereas the case for PR90 soot is from ~ 14 to ~6 nm and with a smaller cumulative pore volume change. This results from the difference in soot sample original primary particle network. Likely, PR90 soot, comprising of a small fraction of volatile content (~ 1%), may have a tight primary particle connectivity. Compared to PR90 soot, FW200 soot may have looser and less intimate primary particle connections, providing a higher probability for primary particle rearrangement when the physical force is imposed. For example, Marcolli et al. (2021) used the concept of overlap coefficient for primary particles to depict the tightness of primary particle connection which has an important influence on the mesopore sizes of soot aggregates. Furthermore, the pore structure in FW200 soot aggregates is not as stable as that in PR90 soot aggregates also resulting from its lower carbonization (Table 1). Therefore, agitation induced aggregate compaction generate more smaller mesopores for FW200 soot than for PR90 soot. Furthermore, compacted PR90 soot has a comparable amount of mesopore structures in the size range ~ 6-7 nm to that of fresh FW200 soot, both showing a



volume based PSD value of ~ 1,000 $mm^3g^{-1}nm^{-1}$ in this pore size range. This explains the comparable IN ability of 400 nm PR90comp and FW200fresh at $T \leq 228$ K. Again, it implies that originally hydrophobic soot can be active INPs if it contains sufficient mesopores relevant to PCF activation. Finally, Fig. 12 also shows a divergence between Ar and $N_2$ PSD results for the same soot sample, suggesting the PSD result is dependent on probe gas nature. As introduced in Sect. 2.3.2, the quadrupole moment of $N_2$ affects the orientation of the molecule during the sorption process (Thommes et al., 2012), thereby influencing

isotherms measured and the PSD results. Nonetheless, these measurements allow concluding the same fact that the physical agitation shifts the PSD mode of fresh soot to small sizes and leads to significant medium mesopore enrichment and IN enhancement for compacted soot.

### 3.4.2 Pore size distribution retrieved from water vapor isotherms

In this section, we present the calculated soot sample PSD results by applying the Kelvin equation based formulation in

Appendix B to corresponding DVS isotherms, assuming different soot-water contact angle values for soot samples. The contact angle ranging from 0º to 180º, and from completely wettable to non-wettable, can be understood as a measure of surface wettability (Pruppacher and Klett, 1997; Lohmann et al., 2016). Reasonable assumptions for FW200 and PR90 soot need to be made by comparing similar water isotherms of the other soot samples with reported soot-water contact angle values in the literature. For instance, Persiantseva et al. (2004) used the sessile drop technique to measure the contact angle of water droplets

on different soot pellets covering a broad range of wettability in conjunction with water vapor isotherms. The authors reported that the water contact angle on the aircraft combustor produced soot surface spans from 60º to 80º. By comparing the isotherm characteristics, their study can serve as a good reference for soot-water contact angle assumptions in this study because we collected similar type of isotherms. Also note that the contact angle of different soot samples can vary in a broad range. For example, Shin et al. (2010) found that hydrophobic graphene contact angle can be up to approximately 91 ± 1º tested by using

a CCD camera. However, Kireeva et al. (2009) reported that the soot-water contact angle can be as low as 28º for hydrophilic black carbon. Moreover, the water contact angle for INPs even with the same material might show a distribution, as a result of surface heterogeneity. For example, both Welti et al. (2012) and Marcolli et al. (2007) found that a distribution of contact angles describes the mineral INPs IN ability better than a single contact angle value arbitrarily used for IN parameterization. Considering the heterogeneity and complex surface properties of soot particles, to ease the discussion, different contact angle

values are assumed to describe averaged soot particle wettability and to simplify data analysis. In this study, three contact angle values (0º, 45º and 75º) for both FW200 and PR90 soot samples was used for the PSD formulation for DVS analysis. According to DVS isotherm analysis in Sect. 3.3, a low contact angle (45º) may fit the FW200 real surface better whereas a higher contact value (75º) for PR90 might be suitable to its more hydrophobic surfaces. Marcolli et al. (2021) used a contact angle mean value 60º for FW200 soot in the PCF-soot framework study, which is also within the contact angle assumption

range in this study. In brief, these three contact angle values can cover the possible contact angle range of these soot samples





and present the role of contact angle in soot-water interaction activities. The corresponding soot PSD results with different soot-water contact angle assumptions are presented in Fig. 13.

The PSD analysis in Fig. 13 demonstrates that small mesopores generated after compaction are responsible for stronger water interaction abilities of compacted soot compared to the fresh soot. PSD curves with the 0º contact angle assumption is used as a perfect case for soot samples and to serve as a PSD benchmark. Assuming a 45º contact angle for FW200 soot, the PSD curve for FW200comp stays above the FW200fresh in the radius range from approximately 2 to 14 nm in Fig. 13a (dotted lines), suggesting compacted FW200 soot contains more PCF relevant mesopores which are required to be smaller than ~ 10 nm at $T$ < HNT (Marcolli, 2014; Marcolli, 2020). PR90comp soot mainly contains pore structures smaller than ~ 17 nm, as

shown by the PSD curve (dashed line) in Fig. 13b calculated with a soot-water contact angle 75º. However, PR90fresh soot show less mesopore abundance in the pore size range from 2 to 9 nm as its PSD curve stays below that of PR90comp soot. It is just the paucity in this pore size range that leads to the poor IN ability of PR90 fresh soot particles. Additionally, the PSD curves retrieved from DVS isotherms showing a monotonic decrease with increasing pore size is different from PSD curves derived from Ar/N$_2$ measurements in Fig. 12 showing a unimodal distribution as a function of pore radius. This is originated

from contact angle dependence of soot-water interaction and is also because of the different probe gas nature.







**Figure 13. The PSD as a function of pore size ($r_p$) for fresh and compacted (a) FW200 and (b) PR90 soot with contact angle assumptions θ = 0°, θ = 45° and θ = 75° indicated by solid, dotted and dashed lines, corresponding to DVS isotherms shown in Fig. 11.**

Overall, PSD results for FW200 soot derived from DVS measurements coincide with the $N_2$ or Ar PSD results to imply that physical agitation induced soot compaction results in the increment of small mesopores required for the PCF activation and thereby promoting compacted soot to be more active INPs. We believe soot PSD results derived from DVS measurements are more relevant to soot particle IN activities as they both are associated with soot-water interaction activities. Particularly, mesopore abundance is a more important predictor for soot particle IN via PCF mechanism. As discussed in Sect. 3.2, both





soot particle IN and compaction induced IN enhancement show size dependence. We suggest that the size dependence is in fact mesopore abundance dependence as a function of aggregate size. In this context, each mesopore can be viewed as a local PCF active site. Only when the mesopore number density reach a critical amount can the aggregate nucleate ice under a PCF favoured $T$ and RH condition. For example, the PSD results in Fig. 13a can explain why 400 nm FW200comp soot AF increases with a larger slope than that of the fresh soot particles at 233 K as presented in Fig. 6. According to soot PCF mechanism

(Marcolli, 2014), cylindrical mesopores with a radius smaller than ~ 9 nm are relevant to PCF at $T = 233$ K and $RH_w < 94$ %, assuming a 45° contact angle for FW200 soot. The PSD results in Figs. 12a and 13a for FW200 soot both suggest that FW200comp soot has more mesopores smaller than 9 nm in comparison with FW200fresh soot.

Similarly, the mesopore increment for compacted PR90 soot (see Fig. 13b) also explains its enhanced water interaction ability

and promoted IN activity. Particularly, the reason for that compaction can promote PR90 soot IN significantly at 228 K but not at 233 K can also be explained by its PSD results. With a soot-water contact angle assumption 75°, PR90 soot show the existence of small mesopores between ~ 2 and ~ 17 nm (Fig. 13b). Though these mesopores are small enough to induce capillary condensation but are too small to provide enough space for the ice embryo to grow out of the pore at 233 K or to nucleate ice in a small volume of water on the short timescale of our experiments (~ 10 s), i.e. the PCF activation is limited by

ice crystal growth at $T = 233$ K or freezing rate. For example, Marcolli et al. (2021) reported that a three-membered-ring pore in a tetrahedral primary particle packing is smaller in pore volume than a four-membered-ring pore in a cubic packing and is more likely to have the ice growth limitation scenario. This can be the possible case for PR90comp soot. In the second possible scenario is the limitation of a low homogeneous freezing rate for supercooled pore water because the supercooled pore water homogeneous freezing rate is both $T$ and the pore water volume dependent. Based on the parameterization for homogeneous

freezing in confined structures (Ickes et al., 2015; Marcolli, 2020), calculation can be made to show some evidence. For example, 400 nm PR90comp soot particles with a 75° water contact angle show onset $S_i = 1.31$ at $T = 228$ K, suggesting its 3 nm mesopore is responsible for pore water capillary condensation. However, assuming a cylindrical pore with a radius 3 nm and a height 400 nm, supercooled pore water homogeneous freezing at 228 K takes ~ 0.1 s whereas takes ~ 25 s at 233 K which is longer than the particle residence time in the HINC chamber in this study. Therefore, PR90comp soot shows a strong

$T$ dependence and a PCF suppression from $T = 228$ to 233 K. Again, linking the PSD results analysed from DVS measurements for FW200 and PR90 soot to the IN results, both low soot-water contact angle and mesopore abundance exert advantages for FW200 soot in PCF activation and contribute to its higher IN effectiveness compared to PR90 soot.

**4 Conclusion and summary**

The effect of soot particle morphology changes on its IN ability has been systematically investigated in this study. The soot

aggregate morphology was modified only by applying physical agitation. Under mixed phase and cirrus cloud $T$ conditions, size selected soot particle IN activities were measured under varying RH conditions at a fixed $T$. Soot aggregate morphological



properties were characterized both online and offline for IN results interpretation, including effective density of size selected soot particles, soot aggregate microscopic TEM images, bulk sample soot-water interaction ability tested by DVS measurements, as well as soot sample PSD analysis based on Ar, $N_2$ and water sorption measurements. Soot aggregate

densification (compaction) caused by physical agitation is shown by a mass increase of size selected soot particles after a certain period of agitation. Further evidence is observed from more compacted soot aggregate microscopic TEM images for these aggregates dispersed by agitated soot samples, demonstrating that the physical agitation can modify soot morphological properties and squeeze soot aggregate by shrinking intra-aggregate structures. The experimental results show that there is a significant IN enhancement for compacted 400 and 200 nm FW200 and compacted 400 nm PR90 soot particles. The size

dependence shows that a larger aggregate size for the same soot sample is more IN active because the larger soot aggregates are more sensitive to the induced aggregate densification causing mesopore enrichment. With decreasing particle size, the physical-agitation-caused mesopore increment plays a limited role in promoting soot IN abilities. The limited IN activity of small soot aggregates ($\leq$ 200 nm) arises from the low probability of PCF relevant mesopores within a small soot aggregate since small aggregates comprised of a limited number of primary particles cannot form sufficient mesopore structures. PCF

activation may require a cluster of mesopores for IN rather than a few solitary mesopores. Importantly, qualitative analysis of DVS isotherms and PSD quantitative calculation based on vapor isotherms (water, Ar and $N_2$) reveal the role of contact and morphology in soot particle PCF IN process. It is demonstrated that compacted soot nucleating ice crystals at lower RH conditions than the fresh soot at the same $T$ results from the enrichment in mesopore population. DVS measurements also show that soot-water contact angle does not change after soot compaction, demonstrating the single importance of soot particle PSD

in its ice nucleation process via PCF mechanism. Compacted FW200 soot active IN ability is attributed to its abundant mesopores and low water contact angle. Notably, 200 and 400 nm compacted PR90 soot with a high contact angle can be competitive INPs to fresh FW200 soot particles at $T <$ HNT, suggesting that the existence of PCF relevant mesopores plays a preliminary role. In conclusion, the PCF IN of these soot particles counts more on the mesopore availability and a lower soot-water contact angle also facilitates the PCF process.


In summary, findings in this study have revealed the implications of soot particles in cirrus cloud formation through a PCF ice nucleation pathway strongly depending on its morphology. Our study suggests that compacted soot particles with abundant mesopore structures and a low contact angle are in favoured for ice activation via PCF at low $T$ and RH conditions and potentially make considerable contributions to cirrus cloud formation competing with liquid droplet homogeneous freezing.

For instance, soot aggregates were detected in contrail residuals and demonstrated to be engaged in contrail evolution and cirrus cloud formation (Twohy and Gandrud, 1998; Petzold et al., 1998). Some aero-engine soot particles residing in cirrus regime act as INPs in a former ice activation can be released with a compacted aggregate structure (Bhandari et al., 2019; Mahrt et al., 2020b). After a such cloud processing scenario, originally fresh and hydrophobic large size soot particles will become more active INPs thereby impacts cirrus cloud formation more significantly. Therefore, this laboratory study implies

the importance of cloud processing effects on the IN activity of soot particles. Equally important, our 60 and 100 nm soot





particle IN activity results suggest that near source aviation soot particles with a particle size distribution smaller than 100 nm (Lobo et al., 2015; Moore et al., 2017; Liati et al., 2019) are potentially PCF relevant comparing to the homogeneous freezing parameterization reported by (Schneider et al., 2021), which also extends the IN data for size selected soot particles to a lower size limit given that the smallest size selected soot particle IN is 100 nm reported in the literature (Mahrt et al., 2018; Nichman et al., 2019). Our study demonstrates that even in the absence of chemical modifications, morphology alone can directly regulate the IN ability of soot particles. In the future, comprehensive soot property characterization with an emphasis on soot PSD is necessitated to evaluate soot IN ability and soot-water contact angle measurement also contributes to a better evaluation.


**Appendix A: SMPS and CPMA measurements for soot samples**

**Raw size distribution of soot samples**

SMPS measurement results show that the agitated particle size distribution curve shifts to smaller size ranges and the peak mode value decreases, comparing to those of the fresh particles. As a function of mobility size, the particle number size distribution curves for fresh and compacted FW200 and PR90 soot samples are presented in Fig. A1. This demonstrates that agitation induced particle physical property change occurs over the entire size range of the fresh particles.

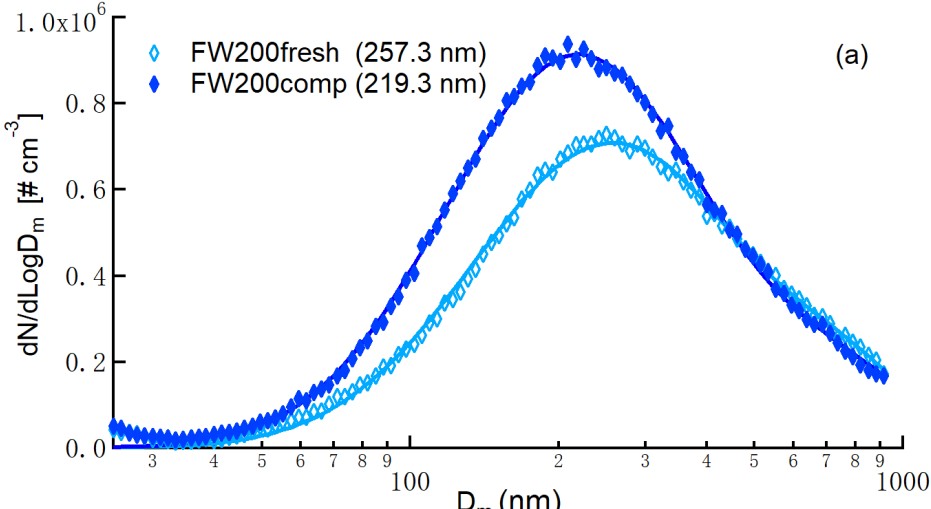



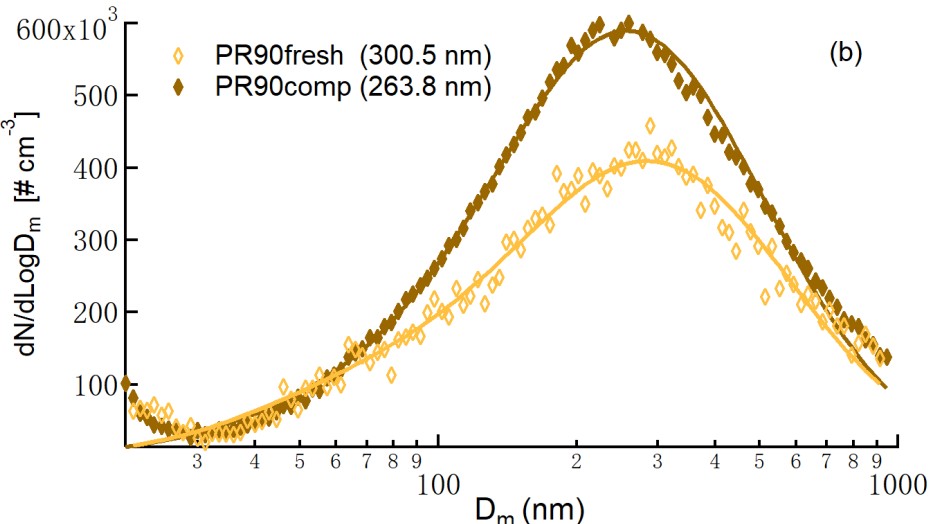

**Figure A1. The size distribution of polydisperse fresh and compacted FW200 (a) and PR90 (b) soot particles as a function of mobility size ($D_m$). Number values in the legend indicate the size distribution mode of the aerosol. The size distribution is shown without multiple charge correction. Solid lines are log normal fitting for the soot aerosol indicated by data points with the same colour.**


**The quality of soot particle size selection**

The DMA selects aerosol particles based on their electrical mobility size, equivalent to the diameter of a perfect spherical particle. Fractal soot particles are not spherical and thus size selected soot aerosol particles deviate from the size selection value and show a biased particle size distribution. We, therefore, measured the particle size distribution of our size selected

soot samples to check the DMA size selection quality (see Figs. A2 to A5). The DMA has a classifier 3080 running with a 3081 column and a polonium radiation source. The SMPS system for particle size distribution measurements has a classifier 3082 and a 3081 column with an X-ray neutralizer, coupling with a CPC 3776 (a low flow mode 0.3 L min$^{-1}$ and a high flow mode 1.5 L min$^{-1}$). The DMA and the SMPS system were operated with the same flow configurations (Table A1) as for all ice nucleation experiments.


**Table A1. The sheath-aerosol flow configuration used for DMA size selection and SMPS size distribution measurements. $D_m$ stands for the size selection value used by DMA. The aerosol to sheath flow ratio in the DMA and SMPS is indicated by $F_{DMA}$ and $F_{SMPS}$, respectively. The flow rate is in volumetric liters per minute.**

| $D_m$ (nm) | $F_{DMA}$ | $F_{SMPS}$ | Size scanning range |
|---|---|---|---|
| 60 | 1 : 18 | 1.5 : 9 | 8–317 nm |
| 100 | 1 : 15 | 1.5 : 7.5 | 9-359 nm |
| 200 | 1 : 12 | 0.3 : 3.0 | 14-594 nm |





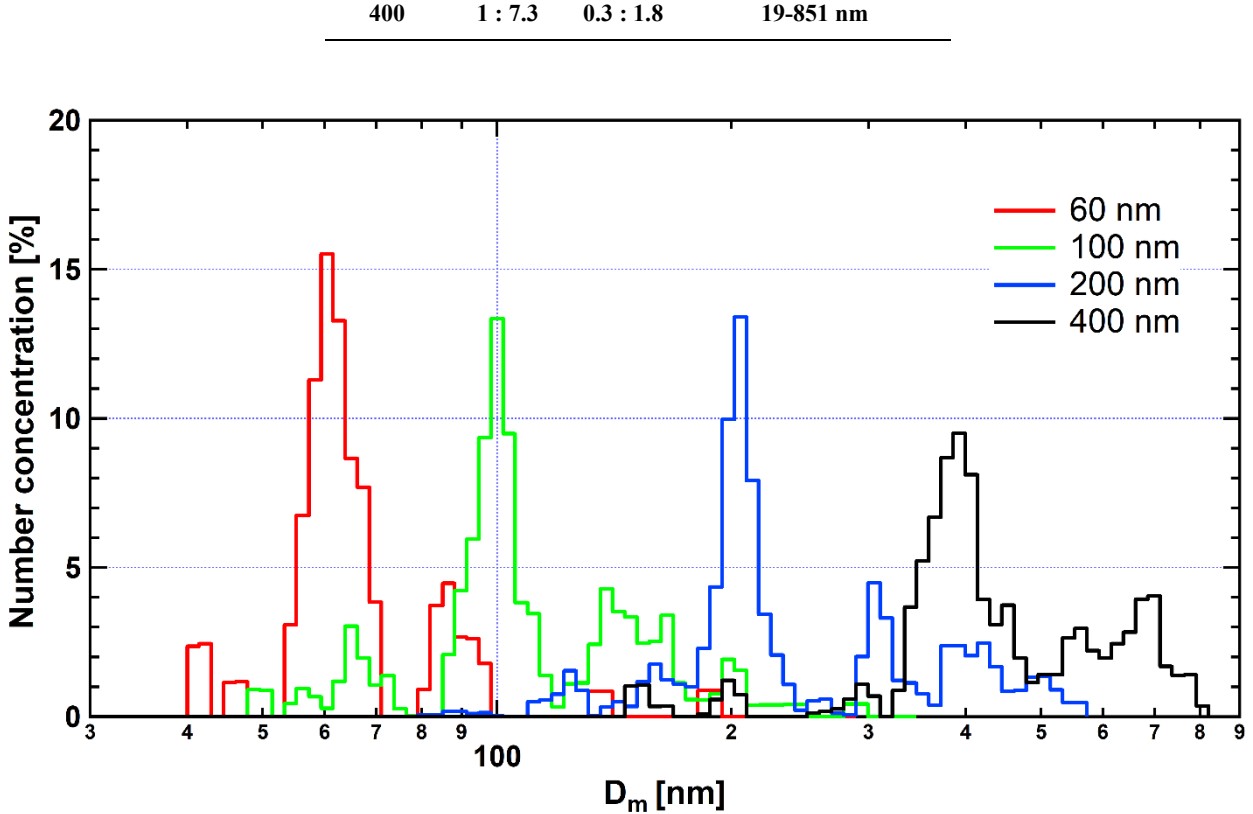

**Figure A2. Number size distribution as a function of mobility size (D_m) for 60, 100, 200 and 400 nm size selected FW200 fresh soot particles.**





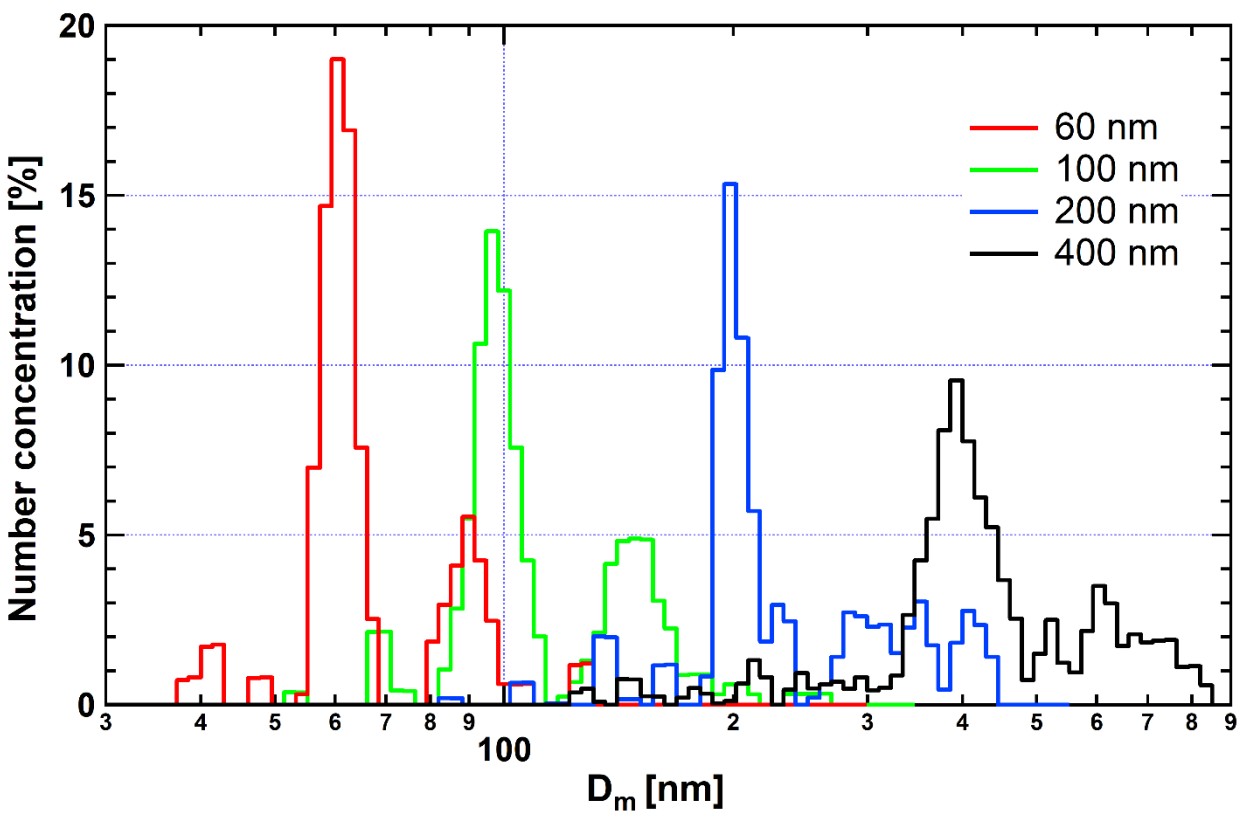

**Figure A3. Number size distribution as a function of mobility size (D$_m$) for 60, 100, 200 and 400 nm size selected FW200 compacted soot particles.**





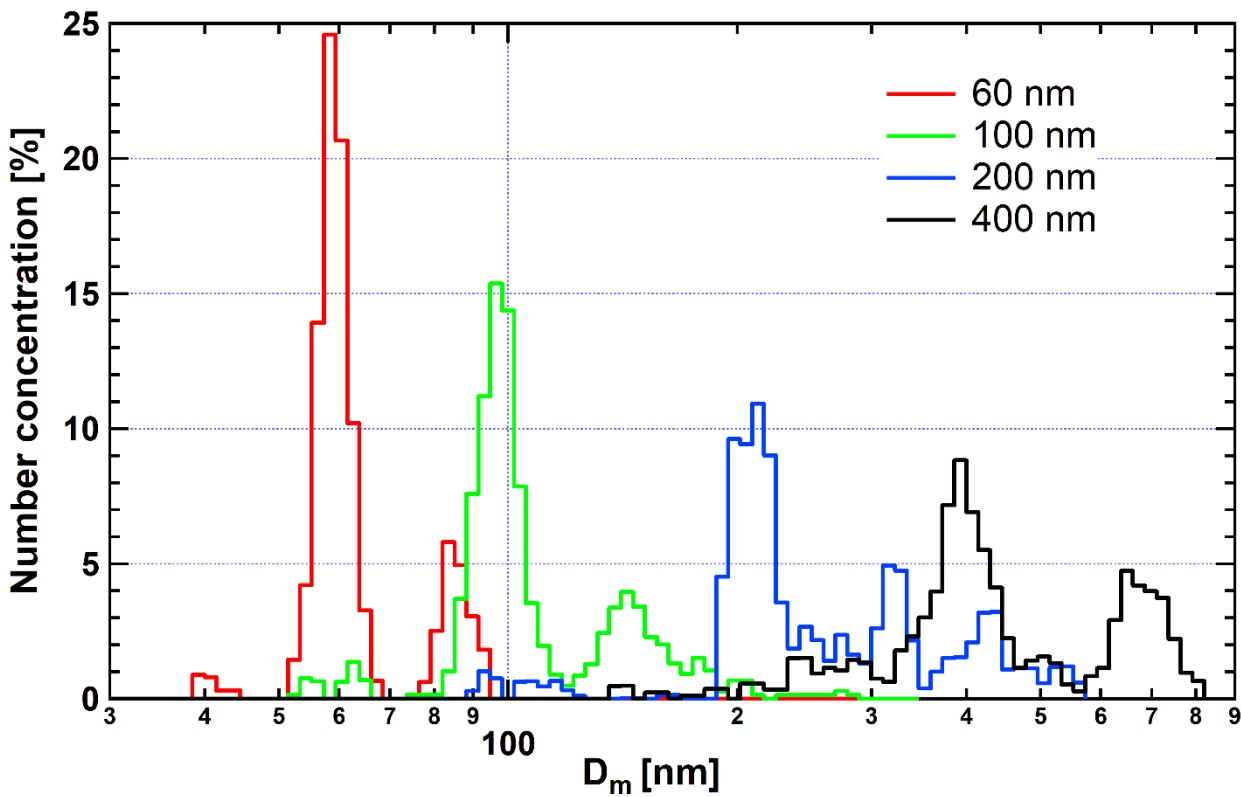

**Figure A4. Number size distribution as a function of mobility size ($D_m$) for 60, 100, 200 and 400 nm size selected PR90 fresh soot particles.**





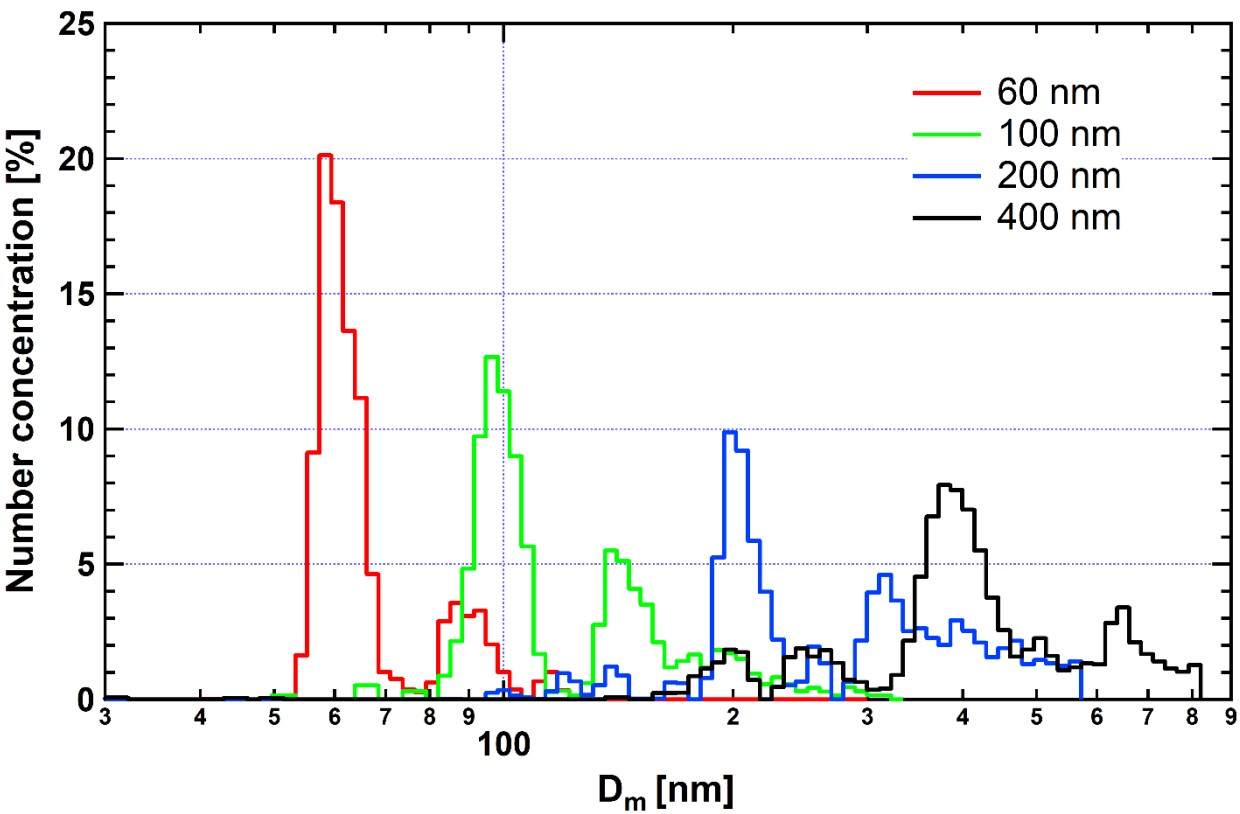

**Figure A5. Number size distribution as a function of mobility size ($D_m$) for 60, 100, 200 and 400 nm size selected PR90 compacted soot particles.**

**Particle fractal dimension calculation**

The fractal dimension $D_f$ (presented in Table 1), describing soot particle morphology, is derived from the power-law relation between particle mobility size and mass (Schmidt-Ott et al., 1990; Olfert and Rogak, 2019).

$$m = CD_m^{D_f} \quad \text{(A1)}$$

where C is the mass-mobility constant.

**Appendix B: Formulation for BET and PSD analysis method based on DVS isotherms**

**BET method formulation**

The BET model is a common approach to estimate specific surface area values, which was developed by Brunauer et al. (1938). Taking gas molecule interactions in adjacent layers into account, Langmuir theory (Langmuir, 1918) of monolayer adsorption





was extended to calculate the surface area of the adsorbent due to adsorption by multiple layers of adsorbates. The BET formula is given by:

$$\frac{1}{V(\frac{p_0}{p}-1)} = \frac{1}{V_m C} + \frac{C-1}{V_m C}\frac{p}{p_0} \quad \text{(B1)}$$

Here, $V$ is the volume of the adsorbate adsorbed by per gram sample at each relative pressure ($p/p_0$) condition, $C$ is a constant and $V_m$ stands for the volume of the adsorbate gas equivalent to a monolayer coverage over the adsorbent. Plotting the raw data as a function of $p/p_0$ results in a linear section of the curve, which is the BET range. The $V_m$ and $C$ values can be easily determined by firstly working out the slope and intercept of the linear function. Finally, the BET specific surface area ($S_{BET}$) can be calculated by the equation below:

$$S_{BET} = \frac{V_m \rho \bar{N} A}{\bar{M}} \quad \text{(B2)}$$

Where $\bar{N}$ is the Avogadro's number $6.02 \times 10^{23}$; $\rho$ is the adsorbate density in gas phase, and $\rho_{N_2} = 0.809 \times 10^{-3}$ g cm$^{-3}$ (Sing, 2014a); $A$ is the cross-sectional area of a probe gas molecule, $A_{N_2} = 0.162$ nm$^2$ (Sing, 2014a); $\bar{M}$ is the molar mass of the adsorbate, $\bar{M}_{N_2} = 28$ g mol$^{-1}$. In this study, the BET range for N$_2$ is from $p/p_0 = 0.05$ to 0.15. The $S_{BET}$ values for N$_2$ physisorption measurements are shown in Table 1.


**PSD formulation for DVS data based on the Kelvin equation**

With a focus on understanding the mesopore PSD, an analysis method based on the Kelvin equation was applied. The Kelvin equation can estimate the critical pore size required for mesopore capillary condensation as a function of water-soot contact angle ($\theta$) and $p/p_0$. The equation is given as Eq. (2) in Sect. 2.3.2. The interfacial tension $\gamma_{sl}$ between the solid and the liquid

phase water is taken as 72 mJ m$^{-2}$ at 298 K (Floriano and Angell, 1990; Hruby et al., 2014). The water molar volume $v_s$ is taken as 18.02 cm$^3$ mol$^{-1}$. According to the Kelvin equation, larger pores require larger $p/p_0$ values than smaller pores to trigger capillary condensation assuming the sample with a uniform $\theta$ distribution on its surface. With decreasing $p/p_0$ level in the desorption process, large pores release adsorbed water first. In this case, the $\Delta m$ at a $p/p_0$ value corresponds to the amount of water desorbed by the pore structures (pore water) whose $r_k$ (see Eq. (2)) are larger than the value calculated by Kelvin

equation at this $p/p_0$ level. The equation to calculate the desorbed water volume or pore water volume is given by:

$$V_W = \frac{\Delta m}{\rho_W} \quad \text{(B3)}$$

Here, $\rho_w$ is the density of bulk water at 298 K, taking 0.99623 g cm$^{-3}$ according to Marcolli (2017). Wheeler (1955) suggested that the amount of water desorbed in the pore is comprised of two parts, including the volume of water freely released from the pore by inverse capillary condensation and the volume of a quasi-liquid layer with a thickness $\delta$ over open surface. Thus,

the radius of a pore ($r_p$) capable of hosting water calculated by Kelvin equation equals to the sum of $r_k$ and $\delta$, given as below:

$$r_p = r_k + \delta \quad \text{(B4)}$$





With decreasing $p/p_0$, $V_w$ increases as a function of decreasing $r_p$. Thus, the relation between $V_w$ and $r_p$ can be derived. Note that the change of $V_w$ in each $p/p_0$ desorption step refers to the sum of the liquid desorption by Kelvin effect and the desorption from the surface of the adsorption film within pores emptied at previous steps. Considering this and assuming a cylindrical pore shape, Shkol'nikov and Sidorova (2007) proposed a PSD function as below:

$$\frac{dV_p}{dr_p} = \left(\frac{r_p}{r_k}\right)^2 \cdot \frac{dV_w}{dr_p} \quad \text{(B5)}$$

Then, the expression for PSD as a function of $r_p$ can be established with assuming a fixed value for $\theta$.

In order to derive the function of PSD having pore size $r_p$ as the variable, the quasi-liquid layer thickness ($\delta$) is still needed. Hence, the following data analysis will be conducted. The DVS raw data is logged as the $\Delta m$ of water adsorbed by the sample to the raw sample weight at varying and discrete $p/p_0$ values. A high order polynomial function was first applied to fit the expression for $\Delta m$ as a function of $p/p_0$ in MATLAB with the coefficient $R^2 > 0.999$. Thus, the approximate function $\Delta m = f(p/p_0)$ was obtained. The mole number of water molecules ($n_w$) adsorbed by per unit mass of the adsorbent can be given as:

$$n_w = \frac{\Delta m}{M_w} \quad \text{(B6)}$$

Where $M_w$, equal to 18.105 g mol⁻¹, is the molar mass of water. Then $n_w$ and $V_w$ can be expressed by $p/p_0$, according to Eqs. (b4) and (b1), respectively. The excess surface work (Adolphs and Setzer, 1996b, a; Churaev et al., 1998), describing the energetic characterisation of adsorption and desorption process, was applied in this study. The Gibbs free energy change ($\Delta G$) during adsorption and desorption process, which describes the thermal dynamic state of the process and is given as Eq. (B7), can be formulated as a function of $n_w$ and $p/p_0$. Essentially, $\Delta G$ is a function of $p/p_0$ and given as:

$$\Delta G = n_W RT \ln\left(\frac{p}{p_0}\right) \quad \text{(B7)}$$

According to Adolphs (2007), there is a critical $p/p_0$ value and a critical value $n_c$ for $\Delta G$ to reach a minimum denoting the water monolayer formation over the sample particle surfaces. The value of $n_c$ can be determined by calculating the $p/p_0$ value for the minimum extremum of $\Delta G$. The thickness ($t$) of the quasi-liquid water layer is also needed to determine $r_p$ and is given as:

$$t = \frac{n_w}{n_c} \tau \quad \text{(B8)}$$

$\tau = 0.24$ nm, is the single water molecule thickness, according to Georgi et al. (Georgi et al., 2017). Finally, given a $p/p_0$, the pore radius ($r_p$) can be calculated by substituting Eqs. (2) (see Sect. 2.3.2) and (B8) into Eq. (B4). By solving simultaneous equations of (B3), (B4) and (B5), the PSD results can be achieved. The corresponding results are presented in Fig. 13 (see Sect. 3.4).





## Appendix C: Activation fraction curves of soot particles with different sizes at $T$ from 243 to 218 K


In Figs. C1 to C4, the AF values derived from 5 μm OPC channel for different soot samples with the same size (400, 200, 100 and 60 nm) are plotted versus RH values at different $T$. The absence of 5 μm OPC signal at $T$ = 243 and 238 K demonstrates that there is no ice nucleation onto soot particles since ice crystals if formed would be able to grow to 5 μm at this temperature. Therefore, the signals in the 1 OPC channel for $T$ = 243 and 238 K are water droplets but not ice crystals. To compare soot

particle IN ability dependence on size, the AF values for the same fresh and compacted FW200 and PR90 soot sample with different sizes as a function of RH at a fixed $T$ are plotted in Figs. C5 to C8, corresponding to a 1 μm OPC channel. Each curve is an average of at least three individual RH scan experiments. Detailed discussions on IN results are presented in the main text in Sect. 3.2.

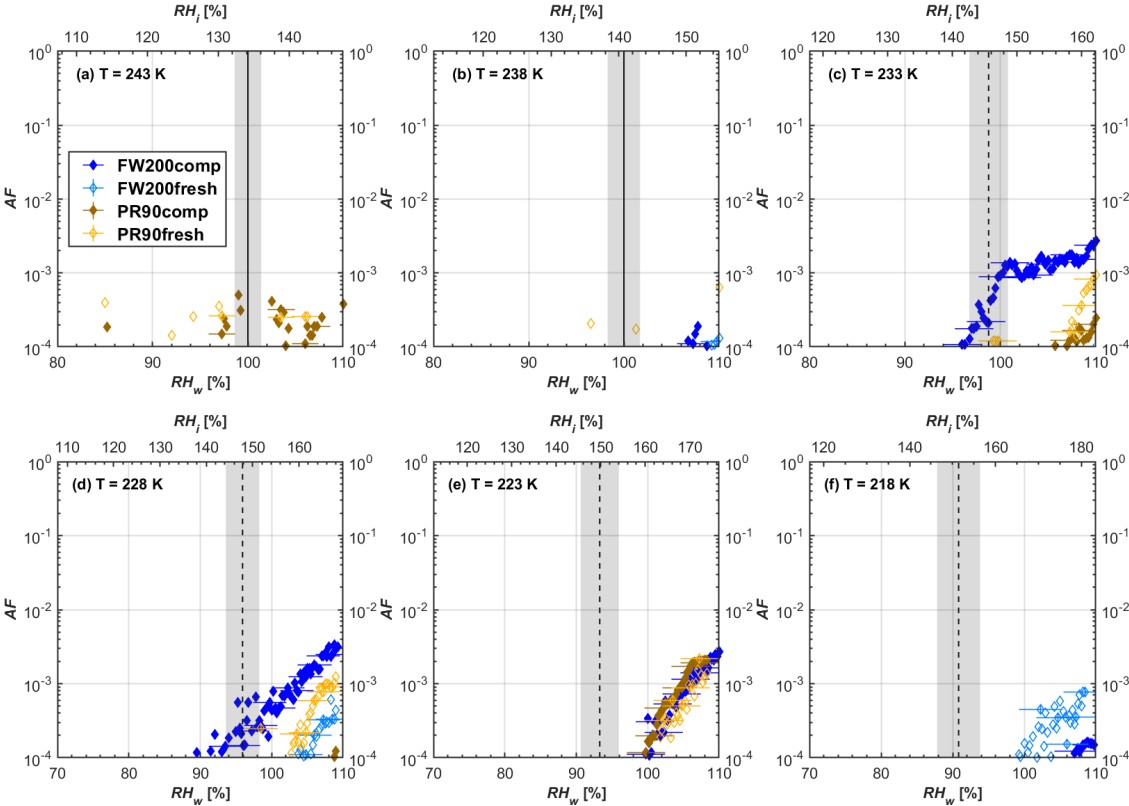

**Figure C1. Averaged AF curves as a function of RH$_w$ and RH$_i$ from the 5 μm OPC channel for 400 nm fresh and compacted FW200 and PR90 soot particles at different $T$. Black solid lines represent water saturation conditions according to Murphy and Koop (2005). Black dashed lines denotes the expected RH values for solution droplet homogeneous freezing at each $T$ (Koop et al., 2000). The grey shading shows the possible variation range in RH that aerosol in HINC can encounter for the calculated homogeneous freezing RH values at each $T$.**





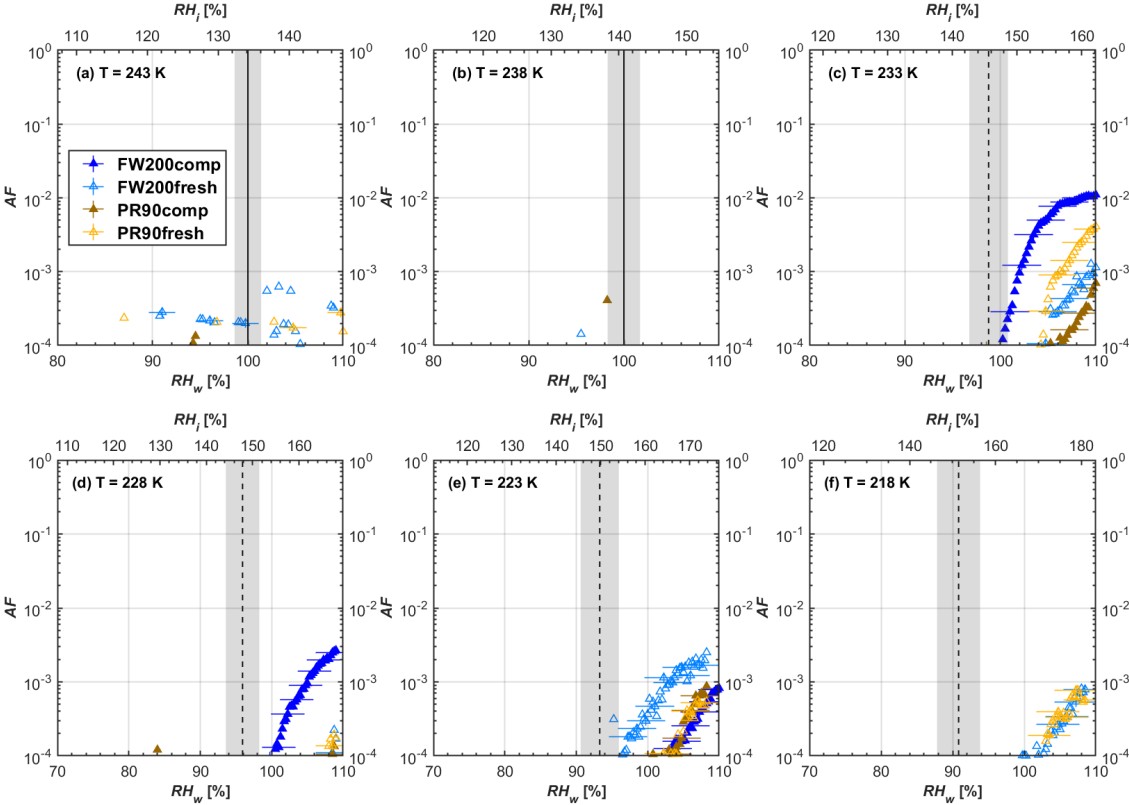

**Figure C2.** Averaged AF curves as a function of RH$_w$ and RH$_i$ from the 5 µm OPC channel for 200 nm fresh and compacted FW200 and PR90 soot particles at different *T*. Black solid lines represent water saturation conditions according to Murphy and Koop (2005). Black dashed lines denotes the expected RH values for solution droplet homogeneous freezing at each *T* (Koop et al., 2000). The grey shading shows the possible variation range in RH that aerosol in HINC can encounter for the calculated homogeneous freezing RH values at each *T*.


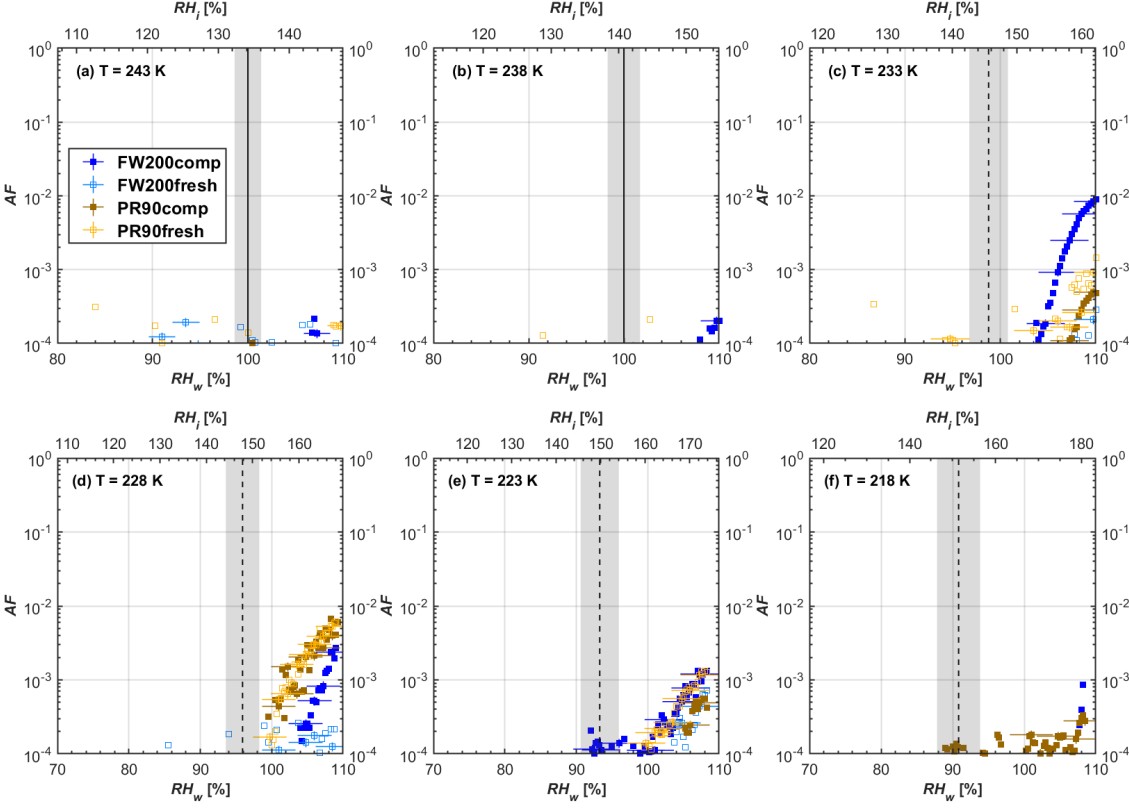

**Figure C3.** Averaged AF curves as a function of RH$_w$ and RH$_i$ from the 5 µm OPC channel for 100 nm fresh and compacted FW200 and PR90 soot particles at different *T*. Black solid lines represent water saturation conditions according to Murphy and Koop (2005). Black dashed lines denotes the expected RH values for solution droplet homogeneous freezing at each *T* (Koop et al., 2000). The grey shading shows the possible variation range in RH that aerosol in HINC can encounter for the calculated homogeneous freezing RH values at each *T*.





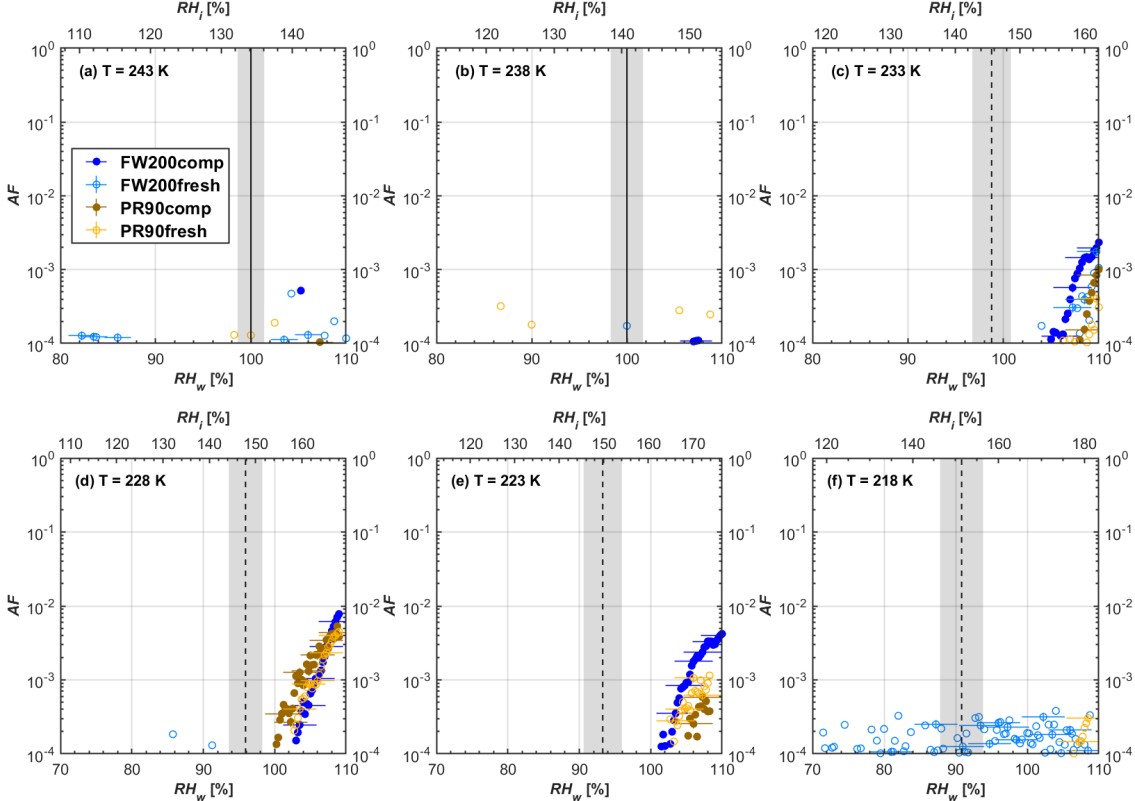

**Figure C4.** Averaged AF curves as a function of $RH_w$ and $RH_i$ from the 5 μm OPC channel for 60 nm fresh and compacted FW200 and PR90 soot particles at different *T*. Black solid lines represent water saturation conditions according to Murphy and Koop (2005). Black dashed lines denotes the expected RH values for solution droplet homogeneous freezing at each *T* (Koop et al., 2000). The grey shading shows the possible variation range in RH that aerosol in HINC can encounter for the calculated homogeneous freezing RH values at each *T*.

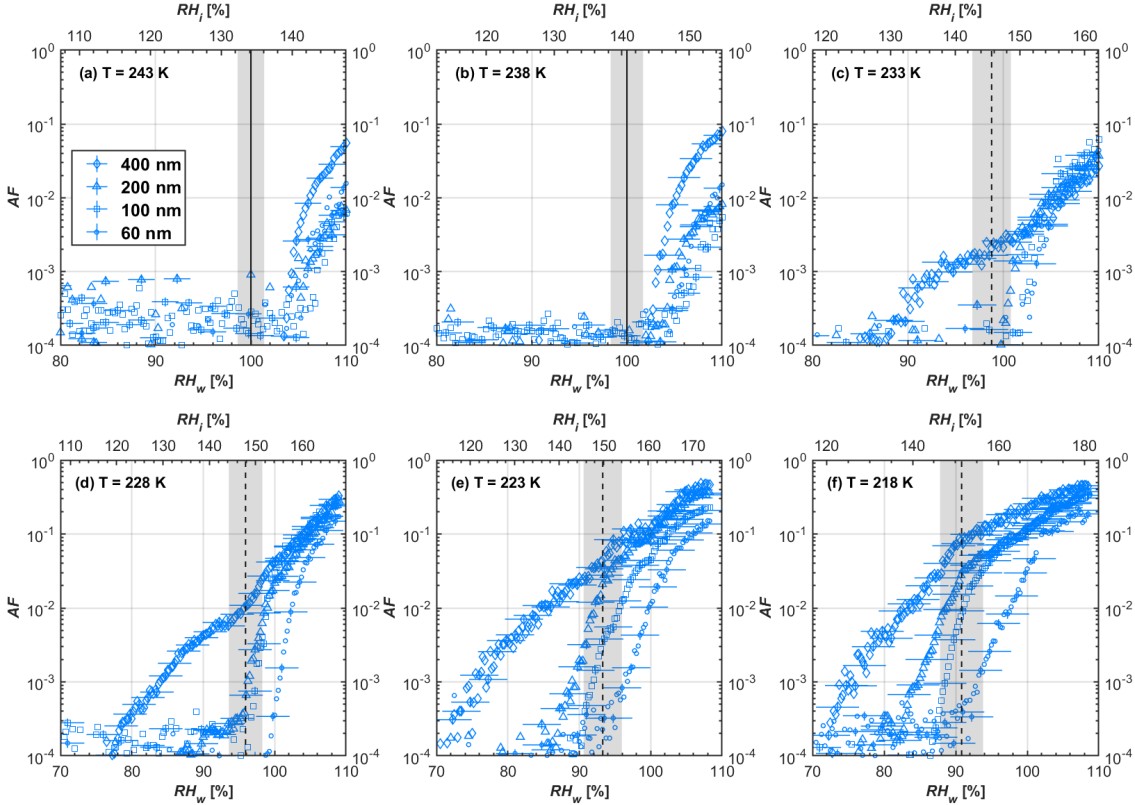

**Figure C5.** Averaged AF curves as a function of $RH_w$ and $RH_i$ from the 1 µm OPC channel for size selected fresh FW200 soot particles at different *T*. Black solid lines represent water saturation conditions according to Murphy and Koop (2005). Black dashed lines denotes the expected RH values for solution droplet homogeneous freezing at each *T* (Koop et al., 2000). The grey shading shows the possible variation range in RH that aerosol in HINC can encounter for the calculated homogeneous freezing RH values at each *T*.



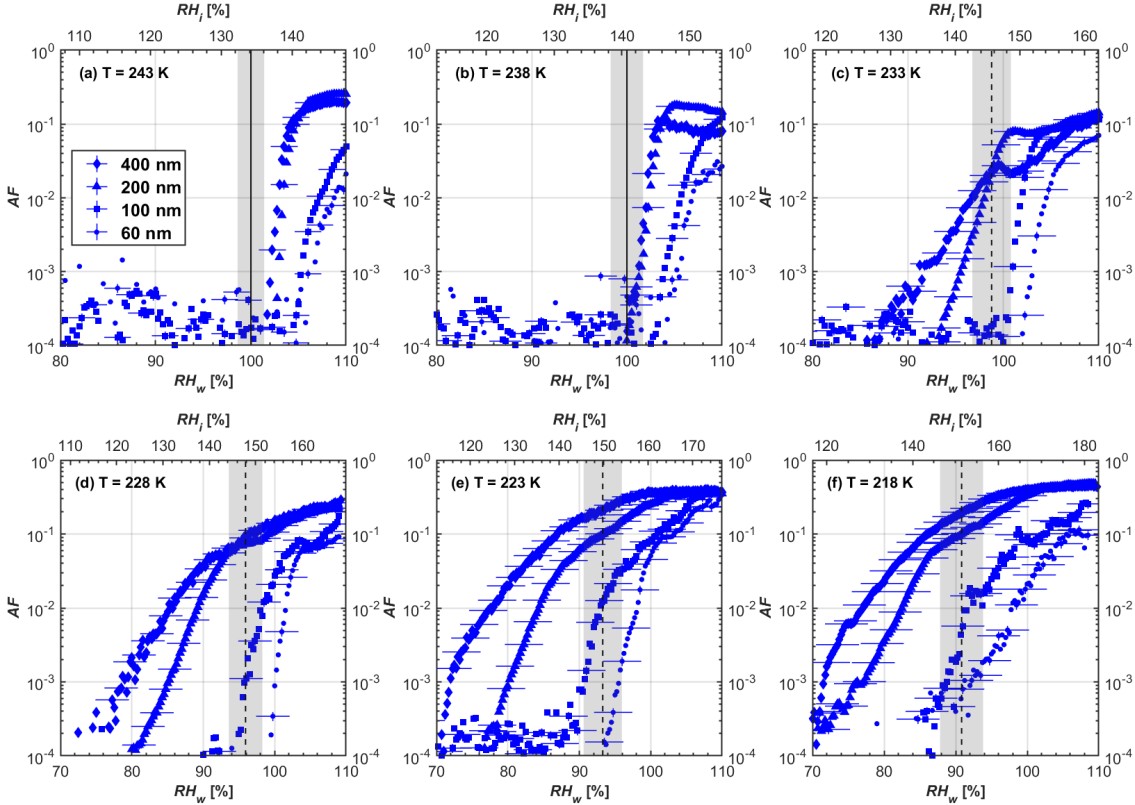

**Figure C6. Averaged AF curves as a function of RH$_w$ and RH$_i$ from the 1 μm OPC channel for size selected compacted FW200 soot particles at different *T*. Black solid lines represent water saturation conditions according to Murphy and Koop (2005). Black dashed lines denotes the expected RH values for solution droplet homogeneous freezing at each *T* (Koop et al., 2000). The grey shading shows the possible variation range in RH that aerosol in HINC can encounter for the calculated homogeneous freezing RH values at each *T*.**




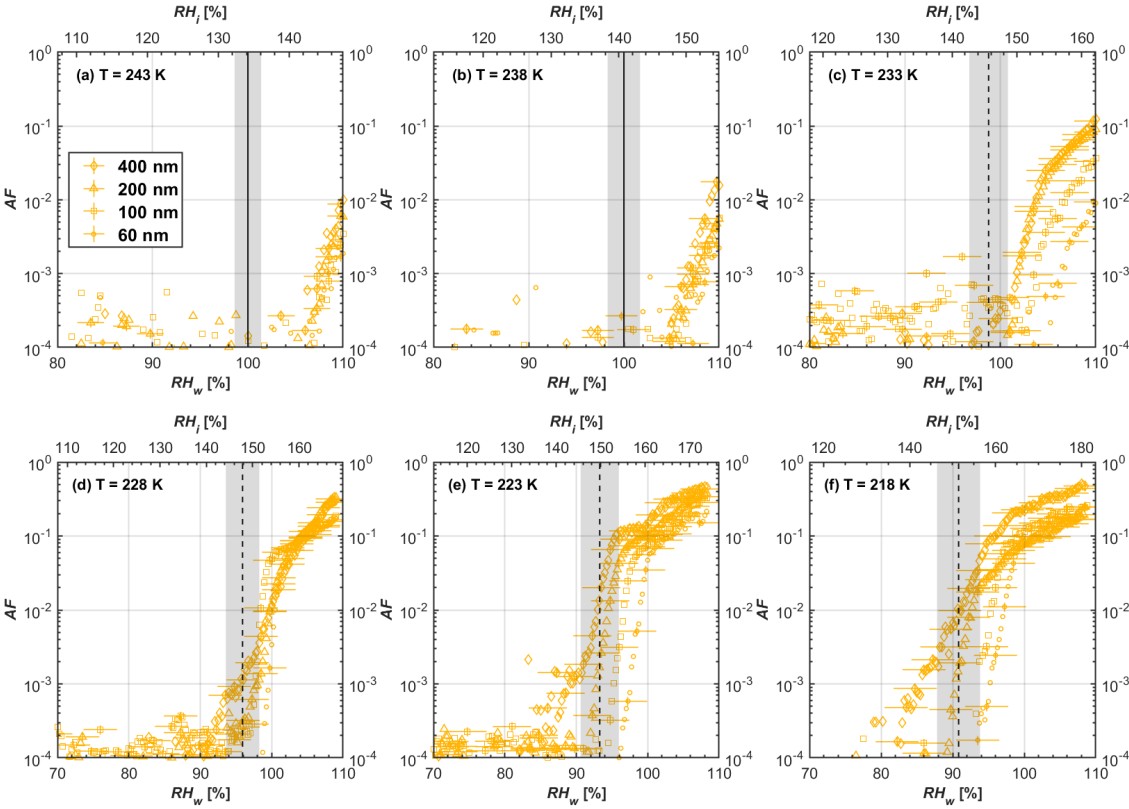

**Figure C7. Averaged AF curves as a function of RH$_w$ and RH$_i$ from the 1 µm OPC channel for size selected fresh PR90 soot particles at different *T*. Black solid lines represent water saturation conditions according to Murphy and Koop (2005). Black dashed lines denotes the expected RH values for solution droplet homogeneous freezing at each *T* (Koop et al., 2000). The grey shading shows the possible variation range in RH that aerosol in HINC can encounter for the calculated homogeneous freezing RH values at each *T*.**




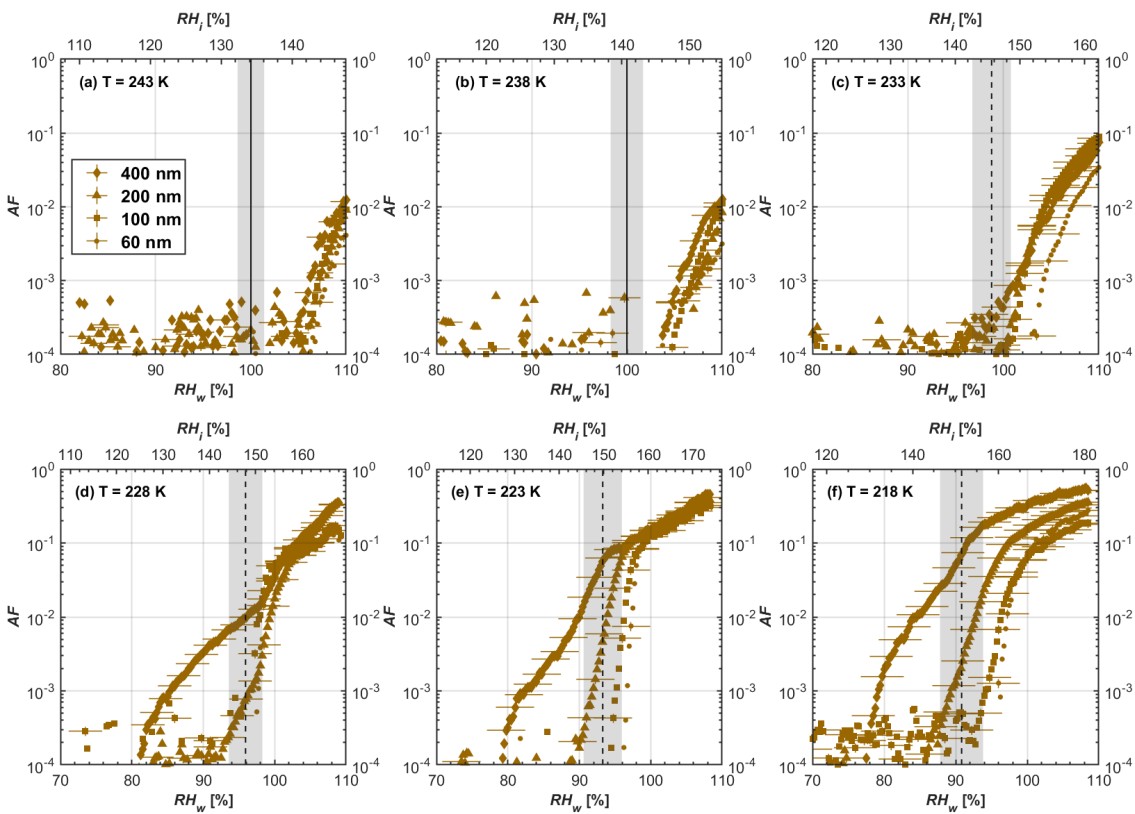

**Figure C8. Averaged AF curves as a function of $RH_w$ and $RH_i$ from the 1 μm OPC channel for size selected compacted PR90 soot particles at different $T$. Black solid lines represent water saturation conditions according to Murphy and Koop (2005). Black dashed**
**lines denotes the expected RH values for solution droplet homogeneous freezing at each $T$ (Koop et al., 2000). The grey shading shows the possible variation range in RH that aerosol in HINC can encounter for the calculated homogeneous freezing RH values at each $T$.**

## Appendix D: Gas sorption isotherms

Ar and $N_2$ adsorption and desorption isotherms are plotted as adsorbed gas volume at the STP (standard temperature and
pressure) condition versus the relative pressure ($p/p_0$) in Figs. D1 and D2, respectively. Both measurements are based on the manometric method which measures the amount of gas removed from the gas phase to evaluate the gas adsorption activity of the soot sample. Ar physisorption measurements are conducted at 87 K and the $N_2$ measurements are performed at 77 K. The $p/p_0$ range is from near zero to approximately unity. Both adsorption and desorption processes are performed. Each curve stands for a single run.





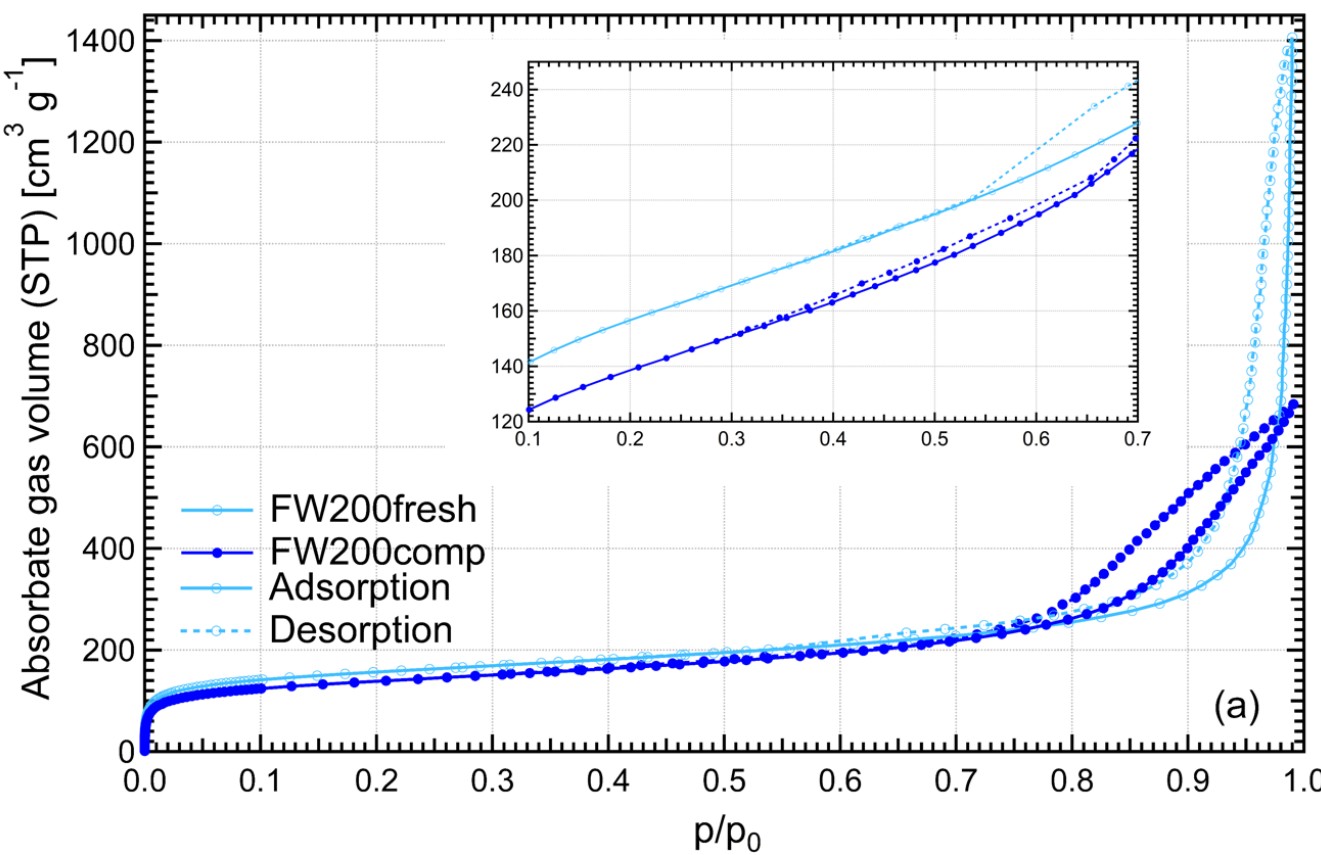





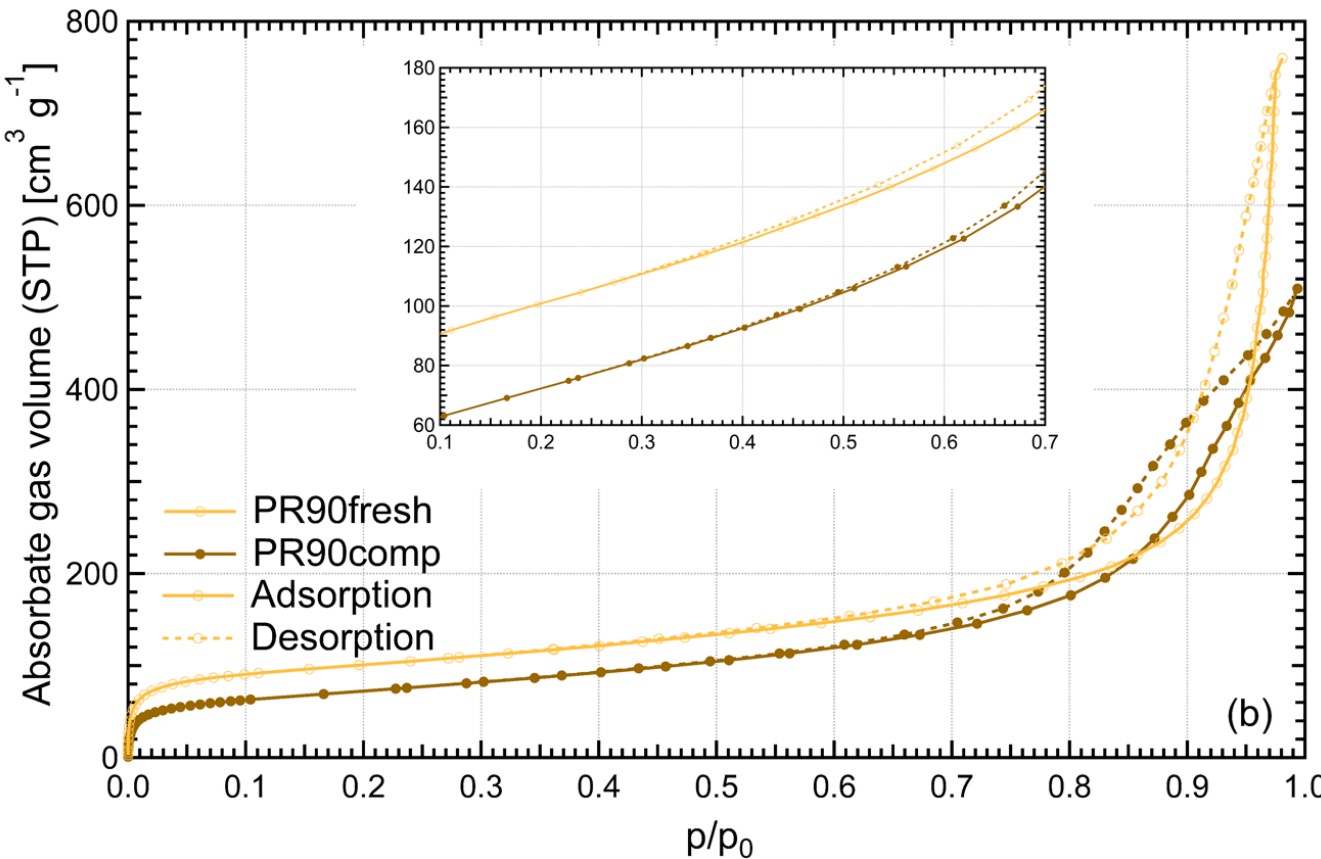

**Figure D1.** Ar adsorption and desorption isotherms plotted as the volume of adsorbed gas for per gram soot sample at standard temperature and pressure (STP) as a function of the relative pressure conditions for fresh and compacted (a) FW200 and (b) PR90 soot. Solid lines indicate adsorption branches and the dashed lines indicate desorption branches.










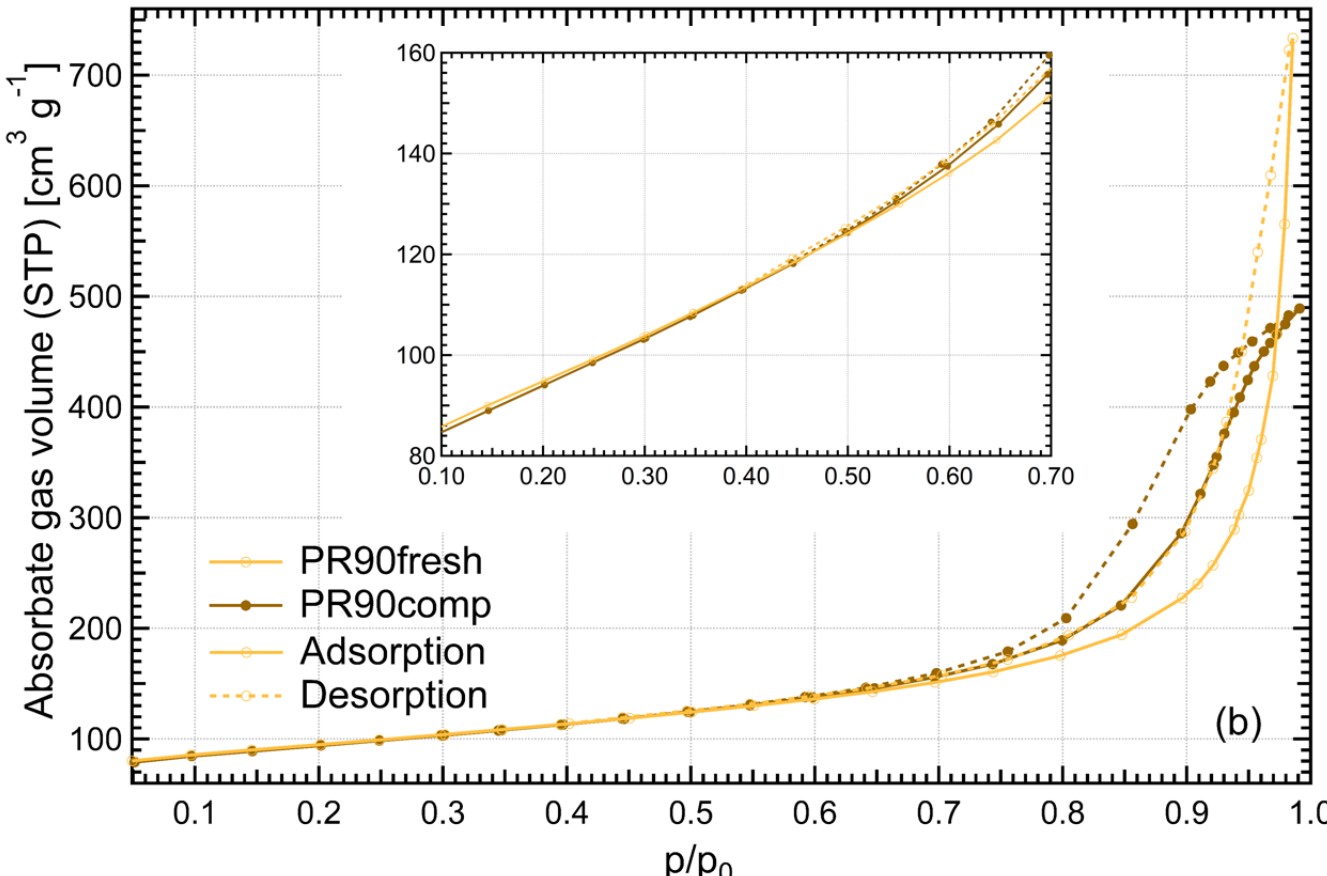

**Figure D2. N$_2$ adsorption and desorption isotherms plotted as the volume of adsorbed gas for per gram soot sample at standard temperature and pressure (STP) as a function of the relative pressure conditions for fresh and compacted (a) FW200 and (b) PR90 soot. Solid lines indicate adsorption branches and the dashed lines indicate desorption branches.**

*Data availability:* The data presented in this publication will be made available at DOI: 10.3929/ethz-b-000511281. Note by the authors: Data and DOI link will be activated for public access upon acceptance of publication.

*Author contributions:* KG and ZAK designed the experiments and interpreted the data. KG conducted the measurements and collected the raw data. KG wrote the manuscript and prepared all the figures with the help from ZAK. FF helped with the analysis of Ar and N$_2$ sorption isotherms using BET and BJH approaches. All authors discussed, reviewed and edited the manuscript. ZAK supervised the project.

*Competing interests:* The authors declare that they have no conflict of interest.





*Acknowledgements:* We are grateful to the experimental atmospheric physics group at ETHZ for their support. This work was supported by Chinese Scholarship Council (Grant No. 201906020041) and the atmospheric physics professorship at ETH. Our thanks also go to Dr. Eszter Judit Barthazy Meier who helped with TEM images visualization. This research was funded by
the Swiss National Science Foundation; SNSF-grant no. 40B1-0_195035.

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
