# Peer review of "Enhanced soot particle ice nucleation ability induced by aggregate compaction and densification"

_Atmospheric Chemistry and Physics, 2021_

## Author Comment (AC1)

We thank Reviewer 1 for their positive and constructive comments on our manuscript ACP-2021-883. In response to the comments, please find our answers and corrections listed below. **Reviewer 1 comments are extracted in bold from original review supplement** and our responses are given directly below in normal font. *The original text in previous manuscript is repeated in red italic* and *corrected text in revised manuscript is typed in blue italic.*

- **General:**

**1    The manuscript presents a comprehensive laboratory investigation of IN properties of soot under cirrus conditions simulated in a continuous-flow diffusion chamber. Two types of commercial soot were analysed fresh and after mechanical agitation, which resulted in compaction of soot particles as revealed by electron microscopy. The resulting four kinds of soot material were further separated into four size classes each and investigated by $N_2$ and Ar sorption measurements regarding differences in their pore size distribution. Soot water interaction was precisely characterised by dynamic vapour sorption measurements. Differences in morphology, in particular mesopore abundance as a factor of compaction and particle size, and the resulting propensity for pore condensation freezing, convincingly explain observed differences in IN activity between the four kinds of soot material and their size classes.**

**The clear structure of the manuscript makes it easy to follow. There is little I can suggest in terms of further improvements, except perhaps a broader outlook on the implications following from these investigations. For example, what might be the effect of future changes in jet fuel composition, such as additions of biofuel or synthetic fuel, to the IN activity of the generated soot?**

R:    Thank you for your positive comments and constructive suggestions.

Regarding adding a broader outlook, we agree with this suggestion and now add some new sentences at the end of Sect. 4 as below (L807-816 in revised manuscript):

'*Finally, synthetic aviation fuels, which contain a negligible amount of aromatics and are gaining an increasing share in aviation fuel usage, can lead to a reduction in aero-engine particulate matter mass and number emissions (Duong et al., 2018; Xue et al., 2019) and significant decrease in the GMD (geometric number mean diameter, short as GMD in L86 in revised manuscript) of aero-engine emitted particle population by up to 40 % (Corporan et al., 2007; Corporan et al., 2011). Considering the IN dependence on soot particle size, it implies that the soot emission from aviation activities fueled by synthetic fuels should have a decreasing impact on cirrus formation. Bräuer et al. (2021) reported that low-aromatic biofuel blends considerably reduce aircraft emitted ice by 40 %. However, Kärcher et al. (2021) noted that a small number of active soot particles in aviation contrails may still modulate cirrus cloud microphysics, by decreasing the ice crystal number concentration but increasing the ice crystal mean size. Hence, further constraining the understanding of the IN activity of small size soot particles is necessary to mitigate aviation soot impacts on the climate.*'

- **Technical issues:**

**1    L22: consider replacing 'agitation degree' by 'degree of compaction'**

R:    Agreed (now L22 in revised manuscript):

'*... the IN activity of soot particles with the same* *degree of compaction, ...*'

**2    Figure 1: What does 'xx #/cc' stand for?**

R:    The 'xx #/cc' stands for the particle number concentration measured the condensation particle counter (CPC) in Figure 1. Now, these 'xx #/cc' labels are removed from Figure 1, to avoid misunderstanding.

**3    L277: replace 'an' with 'a'.**

R:    Yes, it was replaced as below (now L283 in revised manuscript):

'*... in addition to a release of $N_2$ or Ar from pores ...*'

**4    Figure 5: Please say in the legend what the red circles in panels a and b are indicating.**

R:    Thank you for this suggestion. Instead of using a legend to indicate the red circle, we now add a statement in the caption because Panel (a) and (c) in Fig. 5, where the red circles are marked, are busy. Now the caption for Fig. 5 is as below (now L352-353 in revised manuscript):

'*TEM images for 400 nm fresh and compacted FW200 and PR90 soot aggregates at a magnification value 75k. The red circle is used to indicate intra-aggregate voids with size of tens of nano-meters.*'

**5    Line 358: This sentence needs rephrasing. Perhaps in this way: 'However, the rest of the AF curves for small size (60 and 100 nm) are not significantly different for compacted and fresh FW200 or PR90 soot.'**

R:    Thank you for this comment. We agree that this sentence needs to be rephrased but we think the following correction might be better (now L365-366 in revised manuscript):

'*However, the rest of the AF curves for small size (60 and 100 nm) particles do not show significant differences between compacted and fresh FW200 or PR90 soot.*'

**6    Line 411: replace 'shown that' with 'shown by the fact that'**

R:    Yes, it was replaced (now L419 in revised manuscript):

'*...PR90 soot particles are less active INPs compared to FW200 as shown by the fact that the same size fresh or compacted PR90 soot particles...*'

**7    Line 459: do you mean 'still more competitive'?**

R:    Within context of L459 (in original manuscript) in Sect. 3.2.1, we mean soot-aggregate compaction still can play a role in promoting the ice nucleation ability of 200 nm FW200comp particles at 233 K close to the homogeneous freezing temperature. To clarify this, the statement was changed as below (now L466-468 in revised manuscript):

'*Particularly, there is also a reduction in onset $S_i$ value at T = 233 K for 200 nm FW200comp soot particles showing a smaller $S_i$ value compared to 200 nm fresh FW200 soot particles, which means that soot-aggregate compaction can even promote, albeit by a small $S_i$ reduction, 200 nm FW200comp to nucleate ice.*'

The original statement is as following:

'*Particularly, there is also a reduction in onset $S_i$ value at T = 233 K for 200 nm FW200comp soot particles showing a smaller value just out of the error bar compared to 200 nm fresh FW200 soot particles, which means the compaction induced IN enhancement still plays a role, albeit minor at T = 233 K.*'

**8    Line 529: 'the' instead of 'THE'?**

R:    Yes, it was corrected as below (now L538 in revised manuscript):

'*... between adsorptive gas and the solid, ...*'

**9    Line 632: 'too large' instead of 'too larger'?**

R:    Yes, it was corrected as below (now L642 in revised manuscript):

'*These pore structures are too large to trigger inverse Kelvin effects ...*'

**10    Line 773: 'preliminary' or 'primary?**

R:    Here, we mean mesopore structures relevant to the pore condensation and freezing (PCF) process is the prerequisite property for a soot particle to nucleate ice via PCF and is more important than the particle contact angle (surface wettability). Now, we replace the word 'preliminary' with 'prerequisite', which we think it is better than 'preliminary' and 'primary'. Please see the correction as below (now L782-783 in revised manuscript):

'*… suggesting that the existence of PCF relevant mesopores, as a prerequisite for soot PCF activation, is more important than the contact angle.*'

**11    Line 778: replace 'are in favoured for' by 'favour'**

R:    Yes, it is replaced as below (now L788 in revised manuscript):

'*Our study suggests that compacted soot particles with abundant mesopore structures and a low contact angle favour ice activation via PCF at low T and RH conditions ...*'

**12    Lines 781-782: I do not understand the sentence starting with 'Some aero-engine soot particles...'**

R:    Here, we mean aviation soot particles may experience multiple cloud ice activation cycles, i.e. cloud processing (Mahrt et al., 2020b). Now, the sentence is changed as below (now L791-793 in revised manuscript):

'*Some aero-engine soot particles emitted directly in the upper troposphere can act as INPs in a first ice formation cycle and release residuals upon sublimation of this ice. The released residuals can have a compacted aggregate structure (Bhandari et al., 2019; Mahrt et al., 2020b). After such a…*'

**References:**

Bhandari, J., China, S., Chandrakar, K. K., Kinney, G., Cantrell, W., Shaw, R. A., Mazzoleni, L. R., Girotto, G., Sharma, N., Gorkowski, K., Gilardoni, S., Decesari, S., Facchini, M. C., Zanca, N., Pavese, G., Esposito, F., Dubey, M. K., Aiken, A. C., Chakrabarty, R. K., Moosmuller, H., Onasch, T. B., Zaveri, R. A., Scarnato, B. V., Fialho, P., and Mazzoleni, C.: Extensive Soot Compaction by Cloud Processing from Laboratory and Field Observations, Sci. Rep., 9, 11824, https://doi.org/10.1038/s41598-019-48143-y, 2019.

Bräuer, T., Voigt, C., Sauer, D., Kaufmann, S., Hahn, V., Scheibe, M., Schlager, H., Huber, F., Le Clercq, P., Moore, R. H., and Anderson, B. E.: Reduced ice number concentrations in contrails from low-aromatic biofuel blends, Atmos. Chem. Phys., 21, 16817-16826, http://10.5194/acp-21-16817-2021, 2021.

Corporan, E., DeWitt, M. J., Belovich, V., Pawlik, R., Lynch, A. C., Gord, J. R., and Meyer, T. R.: Emissions Characteristics of a Turbine Engine and Research Combustor Burning a Fischer-Tropsch Jet Fuel, Energy & Fuels, 21, 12, 2007.

Corporan, E., Edwards, T., Shafer, L., DeWitt, M. J., Klingshirn, C., Zabarnick, S., West, Z., Striebich, R., Graham, J., and Klein, J.: Chemical, Thermal Stability, Seal Swell, and Emissions Studies of Alternative Jet Fuels, Energy & Fuels, 25, 955-966, http://10.1021/ef101520v, 2011.

Duong, L. H., Reksowardojo, I. K., Soerawidjaja, T. H., Pham, D. N., and Fujita, O.: The sooting tendency of aviation biofuels and jet range paraffins: effects of adding aromatics, carbon chain length of normal paraffins, and fraction of branched paraffins, Combust. Sci. Tech., 190, 1710-1721, http://10.1080/00102202.2018.1468323, 2018.

Kärcher, B., Mahrt, F., and Marcolli, C.: Process-oriented analysis of aircraft soot-cirrus interactions constrains the climate impact of aviation, Commun. Earth Environ., 2, https://10.1038/s43247-021-00175-x, 2021.

Mahrt, F., Kilchhofer, K., Marcolli, C., Grönquist, P., David, R. O., Rösch, M., Lohmann, U., and Kanji, Z. A.: The Impact of Cloud Processing on the Ice Nucleation Abilities of Soot Particles at Cirrus Temperatures, J. Geophys. Res. Atmos., 125, 1-23, https://doi.org/10.1029/2019jd030922, 2020b.

Xue, X., Hui, X., Vannorsdall, P., Singh, P., and Sung, C.-J.: The blending effect on the sooting tendencies of alternative/conventional jet fuel blends in non-premixed flames, Fuel, 237, 648-657, http://10.1016/j.fuel.2018.09.157, 2019.

---

## Author Comment (AC2)

Response to acp-2021-883 reviews for RC2

We thank Reviewer 2 for their effort and positive feedback on our manuscript ACP-2021-883. In response to the comments, please find our answers and corrections listed below. **Reviewer 2 comments are extracted in bold from original review supplement** and our responses are given directly below in normal font. *The original text in previous manuscript is repeated in red italic* and *corrected text in revised manuscript is typed in blue italic.*

**This paper presents results from laboratory experiments investigating the ice nucleating (IN) ability of two types of soot. Test samples were physically agitated to change their physical morphology and the IN ability of the fresh and "aged" particles were measured with a continuous flow diffusion chamber. Samples were also meticulously characterized to determine the physical changes induced by mechanical agitation. The experiments are well designed and extremely thorough and the results are clearly presented. I recommend that this paper be published after consideration of a few minor points listed below.**

1  **Two types are commercially available soot were used for this study. However, there is no discussion of why these soot types were chosen, and whether they are at all representative of atmospheric soot emissions. Further, results show that the two soot types behaved differently when physically aged, indicating that there is some species dependence to the reported results. Some discussion of the generalizability, and limitations, of the results should be included. Also, some discussion should be included about why these two samples were chosen.**

We agree with the reviewer. First, we respond to why FW200 and PR90 are used as our soot samples in this study. Now, few sentences are added in Sect. 2.1 to this effect how they can represent atmospheric soot particles (L132-138 in revised manuscript), see below:

'*FW200 and PR90 samples are used because they are commercially available carbon black products which allow for 1) large sample sizes as was required in this study and 2) the comparison and validation of experiment reproducibility. Further, the primary particle size of FW200 and PR90 are about 13 and 14 nm respectively (see Table 1), both are close to aviation soot particles which have a mean primary particle diameter of 15 nm (Delhaye et al., 2017). Moreover, FW200 and PR90 contain different volatile content of 20 % and 1 % respectively (see Table 1). This difference also makes our samples representative of atmospheric soot particles with varying volatile content caused by ageing processes during their transportation (Li et al., 2018; Ditas et al., 2018).*'

Secondly, the species dependence of compaction induced IN effect was actually addressed in original manuscript at the end of Sect. 3.2.1 (L410-421, now L418-430 in revised manuscript). We attribute their different IN abilities directly to the differences in sample pore size distribution (PSD) and soot-water interaction abilities (see Sect. 3.2.1). For detailed effects of soot-water interaction ability and PSD on soot IN, we discussed the results and presented comparison both between samples of different compaction levels and between different sample types in Sect. 3.3 and 3.4 respectively. As such, we believe the reviewer's concern was already addressed in the original manuscript in the above indicated sections.

Finally, we respond to the concern raised about the generalizability and limitation of the results from FW200 and PR90 soot samples. We presented the implication of the results in this study to soot IN in the real atmosphere, e.g. aviation contrail formation and evolution. As addressed in L780-783 in original manuscript (now L789-795 in revised manuscript), near source aviation soot particles may undergo multiple cloud formation cycles i.e. cloud processing, by which soot particles may be released

back to the atmosphere with a compacted aggregate structure (Bhandari et al., 2019; Mahrt et al., 2020b). Therefore, our results about the aggregate compaction effect on FW200 and PR90 IN can be a general effect for cloud processed (compacted) atmospheric soot particles which will show IN enhancement compared to fresh (uncompacted) atmospheric soot particles. Regarding to the limitation, we add some statements in Sect. 4 as below (L796-800 in revised manuscript):

'*Soot particles, however, may undergo different atmospheric ageing processes simultaneously, such as cloud processing and external material coating, resulting in more complex property changes (Zhang et al., 2008; George et al., 2015; Bhandari et al., 2019) than the single compactness change in this study. Consequently, other property changes induced by atmospheric ageing, such as surface wettability change, should also be considered when evaluating atmospheric soot IN. Furthermore,...*'

**2    The soot samples were "aged" by mechanical agitation with a stir rod for up to two weeks. The authors claim that this resulted in only physical changes to soot particles. However, is it possible that there was some uptake of organic vapors (which are ubiquitous unless working in extremely clean conditions) during this time? And, if so, how might this have affected the results? This is especially important as the soot samples were first stripped of all deposited vapors via heating under vacuum, and thus would have been very susceptible to organic vapor uptake.**

Our soot samples were stored in glass bottles (as shown in Figure 2) with a plastic cap sealed by a rubber O-ring. During the two-week agitation, the glass bottle was closed and sealed. During the later aerosolization, the bottle is under overpressure as connected to a high pressure ultra pure $N_2$ (generated from liquid $N_2$). Strictly, if any, the uptake of organic vapors for our soot sample would only occur when we transfer samples. However, we do not think that such a short exposure to the ambient would have significant influence on the sample. Also, if any, the organic vapor uptake will occur to all our soot samples (both fresh and compacted) which are stored and aerosolized following the same protocol in our laboratory, there should be no difference between our fresh and compacted soot samples of the same carbon black material with respect to organic uptake, which should be negligible we believe. In addition, it is demonstrated that there is no detectable difference in soot-water surface wettability under low relative humidity conditions between the fresh and compacted soot samples (see Sect. 3.1 and Figure 11), suggesting similar density of active sites for water uptake (Persiantseva et al., 2004; Popovicheva et al., 2008a; Popovicheva et al., 2008b). We therefore believe that the compaction level is the only variant for our soot samples generated by the same type of carbon black material. Possible minor impacts from the ambient air if occurred do not lead to property changes between fresh and compacted soot samples significant enough to influence the ice nucleation results.

**3    Related to the above, atmospheric aging almost always involves chemical changes. Thus, while this study provides a detailed examination of the changes in IN ability due to (likely) physical changes, how can these results be applied to the atmosphere? Some discussion of this should also be included.**

See the response to Comment 1 and relevant statements in Sect. 4 (L786-816 in revised manuscript) where we relate the results in this study to the implication of soot ice nucleation impacts on the atmosphere, including the contribution of cloud processed (compacted) soot particles to cirrus cloud formation, the competition between soot pore condensation and freezing and solution droplets homogeneous freezing, and the effect of complex atmospheric ageing processes (both chemical and

physical ageing processes) on soot ice nucleation.

**4    I would recommend rearranging section 3 and dividing it into separate "Results" and "Discussion" sections. Specifically, I would move figures 11 – 13 earlier when discussing the physical changes to the soot particles. Then present the IN results and have a separate section discussing how the physical changes might result in the IN changes and what the implications of these changes are. However, this is just a recommendation and I leave the final decision up to the authors.**

Thank you for your suggestion. We would like to stay with current structure. In Sect. 3.1, the particle effective density results and visualized soot-aggregate images can already show the major difference between our soot samples. As the compaction effects on soot ice nucleation (IN) enhancement is the key point in this study, we would like to illustrate the IN results in Sect. 3.2 earlier other than continuing presenting new soot sample characterization results. Since we can already link the relation between soot compaction and the IN in Sect. 3.2, Sect. 3.3, where Figures 11-13 are presented, provides further evidence on the soot-water interaction ability and the pore size distribution change to support our understanding of soot IN presented in Sect. 3.2. Presenting all the physical changes (Figures 11-13) before addressing the IN results was something we tried initially, but explaining all the results from Figures 11-13 in context of the IN was impossible because we had not yet addressed the IN yet. As such after addressing the IN we had to refer back to the explanations of Figures 11-13, causing a fair bit of back and forth.

**5    In Figure 4 you should change the symbols for fresh samples to open markers to match the other figures.**

Agreed and changed.

**6    While the paper is generally well written, it would benefit from thorough proofreading for English grammar and word usage.**

Thank you for suggesting to improve the language. We proofread the revised manuscript and list the changes as below but we do not duplicate the corrections made following RC 1. We also note that if the paper if accepted for final publication it will undergo proof reading for English Language as well:

1.   L17-19 in revised manuscript now is as below:

[revised manuscript text omitted]